# SUGARCREPE: Fixing Hackable Benchmarks for Vision-Language Compositionality

**Cheng-Yu Hsieh**[1]*, **Jieyu Zhang**[1]*, **Zixian Ma**[1], **Aniruddha Kembhavi**[2], **Ranjay Krishna**[1,2]

[1]University of Washington, [2]Allen Institute for Artificial Intelligence

{cydhsieh,jieyuz2,zixianma,ranjay}@cs.washington.edu, anik@allenai.org

## Abstract

In the last year alone, a surge of new benchmarks to measure *compositional* understanding of vision-language models have permeated the machine learning ecosystem. Given an image, these benchmarks probe a model's ability to identify its associated caption amongst a set of compositional distractors. Surprisingly, we find significant biases in *all* these benchmarks rendering them hackable. This hackability is so dire that blind models with no access to the image outperform state-of-the-art vision-language models. To remedy this rampant vulnerability, we introduce SUGARCREPE, a new benchmark for vision-language compositionality evaluation. We employ large language models, instead of rule-based templates used in previous benchmarks, to generate fluent and sensical hard negatives, and utilize an adversarial refinement mechanism to maximally reduce biases. We re-evaluate state-of-the-art models and recently proposed compositionality inducing strategies, and find that their improvements were hugely overestimated, suggesting that more innovation is needed in this important direction. We release SUGARCREPE and the code for evaluation at: https://github.com/RAIVNLab/sugar-crepe.

## 1 Introduction

Scholars today herald *compositionality* as a fundamental presupposition characterizing both human perception and linguistic processing [10]. Through compositional reasoning, humans can comprehend new scenes and describe those scenes by composing known atoms [19, 17, 3, 9]. For instance, compositionality allows people to differentiate between a photo of "a girl in white facing a man in black" and "a girl in black facing a man in white". For a while now, vision-language research has sought to develop models that can similarly comprehend scenes and express them through compositional language [22, 20, 29, 15].

Given its importance, a surge of new benchmarks have been proposed to evaluate whether vision-language models exhibit compositionality. Recently, Winoground [45], VL-CheckList [53], ARO [49], CREPE [30], and Cola [38] have entered the machine learning zeitgeist. Evaluation is mostly done through an image-to-text retrieval task formulation [53, 49, 30]: by measuring how often models pick the description, "a girl in white facing a man in black" when presented with an image of it, and avoid choosing the incorrect *hard negative* description, "a girl in black facing a man in white".

In this work, we uncover a crucial vulnerability in not just one but all these image-to-text compositionality benchmarks: We find that a *blind* model that never looks at the image, can identify the correct caption and avoid choosing the supposed "hard negatives". This blind model outperforms a wide array of pretrained vision-language models across the suite of benchmarks [36, 18, 14]. We explain this undesired hackability in existing benchmarks by showcasing that there exists a significant distributional gap between the positive and hard negative captions. For instance, in the ARO bench-

---

* The authors contribute equally to this work.

mark [49], human-generated positive captions differ drastically from the hard negative texts generated by randomly shuffling words in the positive captions. As new research has begun to propose methods that claim to improve compositionality on these benchmarks [49, 38], we find it critical to highlight our findings and propose a solution.

We propose a solution to existing hackable benchmarks by introducing SUGARCREPE, a new benchmark to faithfully evaluate compositionality. In curating SUGARCREPE, we identify two main *biases* [2] that result in the distributional gap between positive and hard negatives; and employ mechanisms to fix the shifts. In particular, we find the current procedure in generating hard negatives introduces descriptions that are (1) not plausible and (2) non-fluent. For example, while the caption "olives and grapes on a plate" is a sensical fluent caption, benchmarks often have non-plausible hard negatives like "olives and grapes inside a plate" or simply incomprehensible ones like "right has word another word. There is a words" (see Table 1 for more examples). We mitigate such biases by first leveraging a modern large language model, ChatGPT [32], to generate plausible and natural hard negative texts instead of relying on simple rule-based templates employed by existing benchmarks [30, 49]. Then, we subsample the dataset through an adversarial refinement process to ensure the identified biases are maximally removed by drawing on recent dataset de-biasing work [50, 41, 23]. Taken together, this workflow is where SUGARCREPE derived its name: **S**ynthetic yet **U**nbiased **G**eneration with **A**dversarially **R**efined Compositional **REP**resentation **E**valuation. We qualitatively and quantitatively verify through both human and automatic evaluations that SUGARCREPE effectively fixes these biases.

With SUGARCREPE, we *re*-evaluate recent methods proposed to improve compositionality. Specifically, we focus on one prominent approach that aims to improve compositionality through data augmentation. This method trains models by generating compositional hard negatives and injecting them within a training batch [13, 49]. Unfortunately, we observe that the effectiveness of this simple data augmentation approach is hugely *overestimated* when evaluated on existing benchmarks, leading to limited improvements on SUGARCREPE. Finally, we evaluate a wide variety of 17 pretrained CLIP models [36, 18, 14], and find that current models still lack compositionality. Our results suggest that to improve compositionality, future work may need more innovative techniques.

## 2 Related Work

We situate our paper amongst existing work on vision-language compositionality, and debiasing datasets for model evaluation.

**Evaluating vision-language compositionality.** Recent works have introduced benchmarks to evaluate the compositionality of vision-language models [36]; they find that current models exhibit little compositional understanding [49, 45, 53, 30, 38] despite their remarkable performance on downstream tasks [36, 25, 43, 1, 47, 48, 52]. Models have a hard time discerning between text containing the same words ordered differently [45]. Models also fail to link objects to their attributes, or understand the relationship between objects [53, 49, 38]. Our work finds that many of the benchmarks used to evaluate compositionality have hackable biases; blind models that do not even look at the image outperform state-of-the-art vision-language models.

**Improving vision-language compositionality.** To enhance vision-language models' compositionality, new proposals suggest training strategies that utilize additional data, models, and/or losses [49, 5, 38, 13, 44]. Amongst them, one prominent approach is to explicitly train the models to distinguish hard negatives from the correct captions [49, 13]. While these approaches appear to improve compositionality on benchmarks, it is unclear if these models achieve such improvements by actually acquiring compositional understanding or by exploiting biases in these datasets. We answer this question in our evaluation.

**Debiasing dataset for faithful model evaluation.** Several prior manuscripts have pointed out that biased datasets could lead to an overestimation of models' true capabilities [16]. They have proposed dataset de-biasing methods to enable more faithful model evaluations [39, 50, 41, 23, 34]. For instance, adversarial filtering [50] iteratively trains an ensemble of classifiers on different training splits and uses them to filter out "easy" negatives for each instance. Building upon adversarial filtering, AFLite [41, 23] filters data instances in a more light-weight manner without retraining a

---

[2] We use biases and artifacts interchangeably in the paper.

Table 1: Existing compositionality benchmarks rely on procedurally-generated hard negatives which often do not make logical sense or are not fluent due to grammatical errors.

| Dataset | Nonsensical Hard Negatives | Non-fluent Hard Negatives |
|---|---|---|
| CREPE [30] | Olives and grape inside a plate.
Ground in a basket on the flowers.
A hair wearing a necklace, with her lady on a table. | A door with panes not in a room; the door has windows.
Right has word another word. There is a words.
A shelf with books in something. There is no background. |
| ARO [49] | The grass is eating the horse.
A gray bathtub is looking at a white cat.
Green ball with a remarkable chair behind a blue scene. | At brown cat a in looking a gray dog sitting is and white bathtub.
Scene with remarkable a ball blue a green behind chair.
Books the looking at people are. |
| VL-CheckList [53] | Sheep is hardwood.
Empty zebras.
The bush speaking in the garden. | An man fishing a food from a wrapper using a paw at a open.
It heaving at a city.
An grouping subduing at a room access. |

model at each iteration and leads to benchmarks that more accurately represent the underlying tasks. We use adversarial refinement to remove biases that creep into the generation of compositionality benchmarks.

## 3 Limit and biases of current compositionality benchmarks

A majority of existing compositionality benchmarks for vision-language models formulate the evaluation task as image-to-text retrieval [53, 49, 30]. We focus on these benchmarks and discuss others [45, 38] in Appendix B. Given an image, the model is probed to select text that correctly describes the image from a pool of candidates. Unlike standard retrieval tasks where the negative (incorrect) candidates differ a lot from the *positive* (correct) text, compositionality benchmarks intentionally design *hard negative* texts that differ minimally from the positive text, in order to test whether the model understands the fine-grained atomic concepts that compose the scene.

**Existing hard negative generation process introduces undesirable biases.** Existing benchmarks generate hard negative texts through rule-based programmatic procedures [53, 49, 30], which produce hard negatives by replacing a word of specific type (an object, attribute, or relation) in the original text, by swapping two words, or by shuffling the word order. We find that such procedures introduce unintentional biases in the generated hard negatives (see Table 1); specifically, we observe two major types of undesirable artifacts: (1) *nonsensical* artifacts, and (2) *non-fluent* artifacts. In order to quantitatively measure these biases, we utilize Vera [27], a plausibility estimation model, to characterize the nonsensical bias. Specifically, we define $\mathrm{Vera}(T)$ to be the plausibility score of a caption $T$, where a higher score suggests more sensical the caption is. Similarly, to capture the non-fluent bias, we leverage a grammar-check model [31] that assigns high scores, $\mathrm{Grammar}(T)$, to more grammatically correct texts. In Figure 1, we find that Vera and the grammar model assign higher scores to positive texts, suggesting that many hard negatives are nonsensical and not fluent.

**Dataset biases render current compositionality benchmarks ineffective.** Given the heavily-skewed score gaps, we show that blind models (*i.e.*, Vera and the grammar model) that simply select the higher-scoring texts as positives and admittedly do not possess any vision-language compositionality, can achieve state-of-the-art performances on existing benchmarks. We compare the the blind models against 17 pretrained CLIP models from three sources: OpenAI's in-house WebImageText dataset [36], LAION [42], and Datacomp [14]. We plot the performances of the blind models and the best-performing CLIP models from each category (Figure 2). Blind models achieves state-of-the-art performances on 9 out of 10 existing benchmark tasks. We provide full evaluation results in Appendix D.1.

## 4 SUGARCREPE

We introduce SUGARCREPE, a new benchmark for faithful evaluation of vision-language models' compositionality based on the image-text pairs of COCO [26]. SUGARCREPE presents two key contributions over existing benchmarks: (1) it drastically reduces the two identified dataset biases (Sec. 4.1), and (2) it covers a broad range of fine-grained types of hard negatives (Sec. 4.2). We present a summary comparison on compositionality benchmarks in Appendix B.

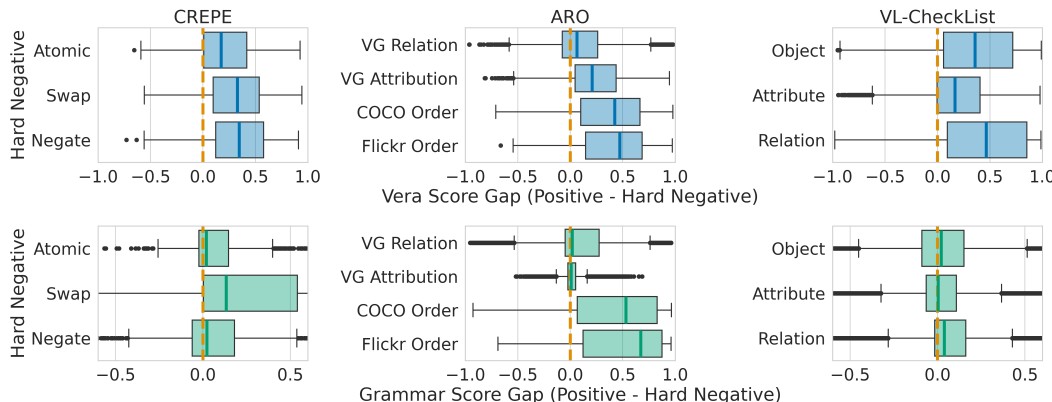

Figure 1: Top row: We define *Vera score gap* as the score difference between the positive and hard negative texts: $\text{Vera}(T^{\text{p}}) - \text{Vera}(T^{\text{n}})$. The entire Vera score gap distribution lies on the positive spectrum, indicating that the template-generated hard negative texts usually have low plausibility. Bottom row: Similarly, *Grammar score gap* is defined by: $\text{Grammar}(T^{\text{p}}) - \text{Grammar}(T^{\text{n}})$. On grammar score, we also find that the distribution largely rests on the positive side, suggesting that most hard negative texts in existing benchmarks exhibit grammatical errors.

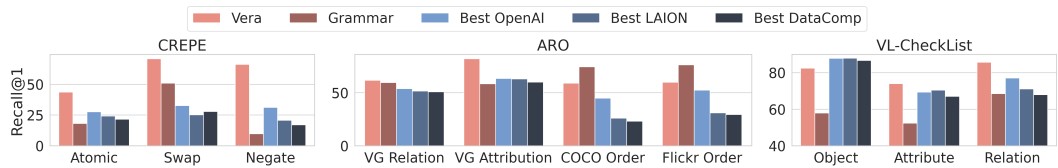

Figure 2: Blind commonsense Vera model and Grammar model outperform state-of-the-art CLIP models on nearly *all* existing benchmarks by exploiting the nonsensical and non-fluent artifacts. This suggests that existing benchmarks are hackable and ineffective in measuring compositionality.

## 4.1 SUGARCREPE generation workflow alleviates dataset biases

The generation procedure of SUGARCREPE consists of three main stages, centered around creating sensical and fluent hard negatives that close the distributional gaps to the positive texts, and ensuring a balanced distribution on the score gaps to make the final dataset robust to the identified biases.

**Stage 1: Generate sensical and fluent hard negatives with a large language model.** Observing the capability of modern large language models in generating fluent and plausible texts, we leverage ChatGPT [32] to generate hard negative texts where we explicitly instruct it to avoid commonsense (logical) and fluency (grammatical) errors. To guide ChatGPT in re-writing a given positive text into its hard negative counterparts, we provide few-shot demonstrations written by the authors and leverage its in-context learning ability [4] to generalize to unseen texts. Figure 3 shows an example demonstration used and an actual hard negative generated. We detail all the prompt templates in Appendix C.2. Table 3 shows the comparisons between hard negatives generated from ChatGPT in SUGARCREPE and that from existing benchmarks.

**Stage 2: Filter false negatives with human validation.** A generated text is considered a valid hard negative only if it incorrectly describes the corresponding image. For example, given an image with a positive caption "a man and a child sitting on a sofa", a compositional change that replaces "child" with "girl" may still result in a correct caption. To ensure the validity of the hard negatives in SUGARCREPE, we filter out *false* negatives by manually examining the generated hard negatives and their corresponding images.

**Stage 3: De-bias dataset with adversarial refinement.** While ChatGPT yields more sensical and fluent text, there is no guarantee that the bias between positive and negative texts is negligible. Following dataset de-biasing work [50, 41, 23], we develop an adversarial refinement mechanism that maximally reduces the undesirably exploitable artifacts in SUGARCREPE. Specifically, our goal is to ensure that performance improvements on SUGARCREPE cannot be achieved by exploiting the

```
Given an input sentence describing a scene, your task
is to:
1. Locate the noun words in the sentence.
2. Randomly pick one noun word.
3. Replace the selected noun word with a new noun word
to make a new sentence.

The new sentence must meet the following three
requirements:
1. The new sentence must be describing a scene that is
as different as possible from the original scene.
2. The new sentence must be fluent and grammatically
correct.
3. The new sentence must make logical sense.

Here are some examples:

Original sentence: A man is in a kitchen making pizzas.
Nouns: ["man", "kitchen", "pizzas"]
Selected noun: man
New noun: woman
New sentence: A woman is in a kitchen making pizzas.

Original sentence: a woman seated on wall and birds
besides her
Nouns: ['woman', 'wall', 'birds']
Selected noun: wall
New noun: bench
New sentence: A woman seated on a bench and birds
besides her.
```

Figure 3: Example prompt (black) and actual hard negative (green) generated from ChatGPT.

**Algorithm 1** Adversarial Refinement

**Require:** Text-only model $M_1$ and $M_2$; Number of grids $K$; A set of candidates $\mathcal{D} = \{I_i, T_i^{\mathrm{p}}, T_i^{\mathrm{n}}\}_{i \in [N]}$, where $I_i$, $T_i^{\mathrm{p}}$, and $T_i^{\mathrm{n}}$ are $i$-th image, positive caption, and negative caption.

**Ensure:** A subset $\bar{\mathcal{D}} \subset \mathcal{D}$

1: Calculate the model score gap for each candidate $g_i^{(1)} = M_1(T_i^{\mathrm{p}}) - M_1(T_i^{\mathrm{n}})$ and $g_i^{(2)} = M_2(T_i^{\mathrm{p}}) - M_2(T_i^{\mathrm{n}})$
2: Split the 2D space $[-1, 1] \times [-1, 1]$ to $K \times K$ equal-size grids.
3: Place each candidate to a grid based on the score gaps $g_i^{(1)}$ and $g_i^{(2)}$.
4: Initialize $\bar{\mathcal{D}} = \{\}$
5: **for** each pair of grid $(G_j, G_j^*)$ symmetric about the original point $(0, 0)$ **do**
6:     **if** $|G_j| > |G_j^*|$ **then**
7:         Sample $|G_j^*|$ candidates from $G_j$ and put them to $\bar{\mathcal{D}}$.
8:         Put candidates in $G_j^*$ to $\bar{\mathcal{D}}$.
9:     **else**
10:         Sample $|G_j|$ candidates from $G_j^*$ and put them to $\bar{\mathcal{D}}$.
11:         Put candidates in $G_j$ to $\bar{\mathcal{D}}$.

identified nonsensical and non-fluent biases. To accomplish this, we characterize the biases again with the commonsense and grammar models [27, 31], and subsample the dataset to ensure symmetric score gap distributions on both the positive and negative sides, as shown in Figure 4. We note the symmetry around zero implies that the commonsense and grammar scores can no longer be used to infer the ground truth positive texts. We provide the adversarial refinement algorithm in Algorithm 1.

### 4.2 SUGARCREPE covers a broad range of hard negative types

To test different aspects of vision-language models' compositional understanding, we follow CREPE [30] to consider various *forms* of hard negatives, and follow VL-CheckList [53] and ARO [49] to consider different fine-grained *categories* of the atomic concepts. In total, SUGARCREPE covers 7 fine-grained types of hard negatives, as shown in Table 2. We introduce the dataset taxonomy below, starting from the *form* of the hard negatives to its different *finer-grained* variants.

**The REPLACE form.** Given a positive text describing a scene, we generate a REPLACE hard negative by replacing an atomic concept in the original text with a new concept that makes the text mismatch with the original scene. Based on the type of the atomic concept—object, attribute, or relation—we further categorize REPLACE hard negatives into REPLACE-OBJ, REPLACE-ATT, and REPLACE-REL.

**The SWAP form.** Different from REPLACE, SWAP does not introduce new concepts in the hard negatives, but a SWAP hard negative is generated by swapping two atomic concepts of the same category in the positive text. We further categorize SWAP into SWAP-OBJ and SWAP-ATT, and omit swapping two relationships since it generally results in nonsensical texts.

**The ADD form.** Similar to the REPLACE form, but instead of replacing an atomic concept with a new one, we generate an ADD hard negative by adding a new atomic concept to the positive text that makes it mismatch with the original scene. We only further categorize ADD into ADD-OBJ (adding object concept) and ADD-ATT (adding attribute concept), as adding new relationship concepts to the positive texts often make them highly implausible.

**Dataset overview.** The final evaluation set of SUGARCREPE consists of $7,512$ examples, where the numbers for each fine-grained type are listed in Table 2. Each example is an image-to-text retrieval task composed of an image, a positive text, and a hard negative. On SUGARCREPE, random chance performance has an average accuracy of $50\%$. We note that ARO and CREPE additionally consider SHUFFLE (randomly shuffling words in a sentence) and NEGATE (adding negation keywords "no/not"

Table 2: We report the number of hard negative captions of all types in SUGARCREPE.

| | REPLACE | | | SWAP | | ADD | |
|---|---|---|---|---|---|---|---|
| | **Object** | **Attribute** | **Relation** | **Object** | **Attribute** | **Object** | **Attribute** |
| **# negative captions** | 1,652 | 788 | 1,406 | 246 | 666 | 2,062 | 692 |

Table 3: We present example positive texts and their hard negatives in ARO+CREPE (generated using existing procedures) and SUGARCREPE (generated with ChatGPT). SUGARCREPE brings significant improvements in commonsense and fluency.

| Hard-Negative Type | Text Type | Commonsense | Fluency |
|---|---|---|---|
| REPLACE | Original | Two adult bears play fight in the water. | A man sitting in front of a laptop computer. |
| | ARO+CREPE | Two adult bears play fight in the soda. | A man sitting around front of a laptop computer. |
| | SUGARCREPE | A flock of ducks play fight in the water. | A man standing in front of a laptop computer. |
| SWAP | Original | A woman standing behind a fence looking at an elephant. | Man swinging tennis racket while group of people watches. |
| | ARO+CREPE | A fence standing behind a woman looking at an elephant. | Group swinging tennis racket while man of people watches. |
| | SUGARCREPE | An elephant standing behind a fence looking at a woman. | Group of people swinging tennis racket while man watches. |
| NEGATE / ADD | Original | A teddy bear next to a stuffed fish. | A red fire hydrant on a city sidewalk. |
| | ARO+CREPE | A teddy bear next to a stuffed fish. There is no teddy bear. | A red fire not hydrant on a city sidewalk. |
| | SUGARCREPE | A teddy bear and a stuffed fish and a robot toy. | A red fire hydrant and a trash can on a city sidewalk. |

to a sentence) hard negatives. We however omit them in SUGARCREPE as SHUFFLE is very unlikely to be plausible and fluent, and NEGATE introduces irreducible keyword artifacts [30]. [3]

## 5 Evaluations

In this section, we qualitatively and quantitatively compare SUGARCREPE to existing benchmarks (Sec. 5.1), re-evaluate recent methods proposed to improve compositionality of vision-language models (Sec. 5.2), and comprehensively evaluate a wide array of pretrained CLIP models (Sec. 5.3).

To systematically and fairly compare SUGARCREPE with existing benchmarks, we normalize the benchmarks by reproducing their data generation workflow using COCO [26] as in SUGARCREPE. We utilize source code from CREPE [30] to generate REPLACE, SWAP, NEGATE hard negatives and take SHUFFLE hard negatives released in ARO [49]. We refer to this reproduced dataset as ARO+CREPE. In addition, we standardize the evaluation task as retrieving the correct caption from *two* possible choices, *i.e.*, a positive text and a hard negative. This normalization sets the positive texts fixed for all benchmarks, including SUGARCREPE.

### 5.1 SUGARCREPE significantly reduces dataset biases

**SUGARCREPE generates more sensical and fluent hard negatives.** We validate that SUGARCREPE generates higher quality hard negative texts by leveraging ChatGPT than previous rule-based approaches. Qualitatively, in Table 3, we observe that the hard negatives in SUGARCREPE are more sensical and fluent compared to hard negatives in ARO+CREPE. We report human evaluation results in Appendix D.2 that show on an average of 35% of examples, hard negatives in SUGARCREPE have *strictly* higher quality than ARO+CREPE in terms of commonsense and fluency. For instance, on SWAP, humans judge that SUGARCREPE wins 68% over ARO+CREPE and ties on 28% of examples in terms of commonsense. Quantitatively, in Table 4, we compare the commonsense and grammar scores averaged over the hard negative texts in both ARO+CREPE and SUGARCREPE. We see SUGARCREPE has much higher average scores than ARO+CREPE. Additionally, pairwise comparisons show that SUGARCREPE has higher commonsense and grammar scores than ARO+CREPE on 86% of examples on average.

**SUGARCREPE disentangles the identified exploitable biases.** We show that the final SUGARCREPE evaluation set maximally reduces the identified biases that could be exploited undesirably to achieve improvements on a benchmark. Figure 4 visualizes the Vera/Grammar score gap distributions. We compare the distributions between ARO+CREPE and SUGARCREPE (before and after adversarial refinement). First, We see that by leveraging ChatGPT, the hard negative texts in SUGARCREPE already have lower biases than ARO+CREPE before adversarial refinement, *i.e.*, the score gap distribution is more centered around zero. Furthermore, we see that after adversarial refinement, the

---

[3]One can easily infer hard negatives from whether the text contains negation keywords "no/not".

Table 4: We compare the commonsense and grammar scores on hard negatives in ARO+CREPE and SUGARCREPE. We report both their respective average scores and the ratio where SUGARCREPE has higher score than ARO+CREPE in pairwise comparison. Overall, SUGARCREPE has hard negatives with better commonsense and grammar.

| Hard-negative Type | Metric | Average Score | | Pairwise Better Ratio |
| --- | --- | --- | --- | --- |
| | | ARO+CREPE | SUGARCREPE | |
| REPLACE | Commonsense | 37.46 | 50.21 | 77.71 |
| | Grammar | 76.79 | 88.96 | 86.85 |
| SWAP | Commonsense | 23.09 | 41.57 | 78.76 |
| | Grammar | 45.67 | 80.46 | 87.02 |
| NEGATE / ADD | Commonsense | 25.24 | 50.20 | 87.24 |
| | Grammar | 65.09 | 90.07 | 95.03 |

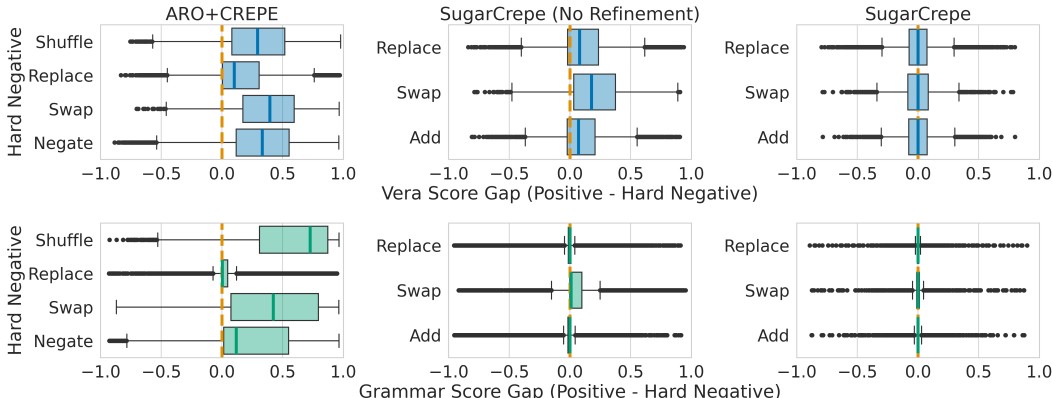

Figure 4: We compare the Vera (top row) and Grammar (bottom row) score gap distributions between ARO+CREPE (leftmost column), SUGARCREPE without adversarial refinement (middle), and SUGARCREPE (rightmost). Top row: We see that Vera score gap distribution shifts from the positive spectrum to more centered around zero from ARO+CREPE to SUGARCREPE without refinement. After adversarial refinement, we ensure the score gap distribution is centered around zero on SUGARCREPE. Bottom row: Similarly, from ARO+CREPE to SUGARCREPE, we see the Grammar score gap distribution shifts from the positive spectrum to centered around zero.

score gap distributions on the final SUGARCREPE evaluation set are symmetric around zero. This implies that the previously identified artifacts can no longer be exploited to infer the positive texts. As a result, we show that the previous commonsense and grammar attacks that are extremely successful on existing benchmarks do not work on SUGARCREPE. As shown in Table 6, these blind models now consistently rank the *last* on SUGARCREPE as compared to other pretrained CLIP models.

## 5.2    Re-evaluating recent methods for improving compositionality

Given the vulnerability of existing compositionality benchmarks, it is unclear whether recently proposed methods that show state-of-the-art performances on these benchmarks are indeed effective. Thus, we re-evaluate these methods with SUGARCREPE.

**Hard negative augmented training.** Specifically, we focus on evaluating one common *data-augmentation* approach considered in [49, 13], where the core idea is to explicitly create hard negatives and train the model to distinguish them. We broadly refer to this training scheme as NEGCLIP following [49]. We evaluate two NEGCLIP training schemes: finetuning and training from scratch. For finetuning, in addition to taking the model released in [49], we finetune another three NEGCLIP models (using ViT-B/32 following [49]) with three respective types of hard negatives (*i.e.*, REPLACE, SWAP, NEGATE) generated using CREPE's [30] source code. For training from scratch, we use RN50 as the base model and train variants of NEGCLIP by augmenting the training examples with different types of hard negatives. We perform both training and finetuning on COCO [26].

**Improvements are overestimated due to unintentionally overfitting.** In Table 5, we first see that NEGCLIP finetuned models show significant improvements on ARO+CREPE, boosting the performance more than 10% compared to standard CLIP finetuning on 11 out of 16 cases (highlighted in green). The lifts are especially large when the hard negative type used in finetuning matches that used in evaluation, where NEGCLIP finetuned models can achieve near human-level performances. For instance, by finetuning with REPLACE hard negatives, NEGCLIP reaches 94% on ARO+CREPE evaluated with REPLACE hard negatives (human performance is 95%). While the results on ARO+CREPE suggest that NEGCLIP is seemingly sufficient in equipping models with strong compositionality, we however see that the improvements brought by NEGCLIP are much smaller on SUGARCREPE. In fact, none of the improvements on SUGARCREPE is larger than 10%, and the best performing NEGCLIP finetuned models still have large gaps to human-level performances, *e.g.*, best NEGCLIP model lags behind human by 23% on SUGARCREPE's SWAP hard negatives. Similarly, when trained from scratch, we observe the same trend that NEGCLIP's improvements are much larger on ARO+CREPE than on SUGARCREPE. The improvements on ARO+CREPE are again most pronounced when the training and testing hard negative type matches.

We attribute the stark contrast in NEGCLIP's effectiveness on ARO+CREPE and SUGARCREPE to model's unintentional overfitting: The NEGCLIP models learned to exploit artifacts that can be used to easily distinguish hard negatives from positives on ARO+CREPE, instead of actually improving compositionality. Thus, when evaluated on SUGARCREPE where the artifacts are removed, the improvement from NEGCLIP drastically reduces. These results imply that NEGCLIP's effectiveness is overestimated on existing benchmarks, and we may still need further innovations to fundamentally improve a model's compositionality. [4]

Table 5: Re-evaluating hard negative augmented training shows that the method's improvements on existing benchmarks (ARO+CREPE) are hugely overestimated, particularly when the test hard negative type matches the one used in training, which can be attributed to overfitting the artifacts.
Color notations: Gains compared to standard CLIP (finetuned / from scratch) > 10% .

| Model | Training | Hard Negative Used | ARO+CREPE | | | | SUGARCREPE | | |
|---|---|---|---|---|---|---|---|---|---|
| | | | REPLACE | SWAP | NEGATE | SHUFFLE | REPLACE | SWAP | ADD |
| Human | | | 95.33 | 100 | 99.33 | 96.00 | 98.67 | 99.50 | 99.00 |
| ViT-B/32 | Pretrained | N/A | 75.71 | 71.58 | 76.89 | 72.06 | 80.76 | 63.27 | 75.09 |
| | CLIP finetuned | N/A | 77.06 | 68.81 | 61.19 | 63.04 | 84.76 | 70.83 | 85.58 |
| | NEGCLIP finetuned | REPLACE | 94.51 | 90.04 | 85.06 | 88.15 | 88.27 | 74.89 | 90.16 |
| | | SWAP | 82.88 | 94.48 | 77.57 | 87.00 | 85.54 | 76.21 | 86.56 |
| | | NEGATE | 77.24 | 68.91 | 99.54 | 64.28 | 84.97 | 70.29 | 85.84 |
| | | Released in [49] | 85.72 | 94.35 | 83.51 | 90.45 | 85.36 | 75.33 | 87.29 |
| RN50 | CLIP from scratch | N/A | 69.93 | 59.96 | 55.36 | 68.78 | 69.54 | 60.33 | 67.63 |
| | NEGCLIP from scratch | REPLACE | 89.04 | 66.51 | 60.90 | 75.23 | 74.32 | 62.65 | 72.92 |
| | | SWAP | 72.33 | 92.29 | 64.51 | 84.84 | 73.31 | 68.35 | 71.93 |
| | | NEGATE | 70.09 | 60.29 | 99.45 | 69.03 | 72.74 | 60.89 | 70.47 |
| | | REP + SW + NEG | 86.30 | 88.60 | 99.34 | 82.93 | 75.26 | 67.69 | 73.08 |

## 5.3 Comprehensive evaluations on existing pretrained vision-language models

We present four key findings in our evaluation over 17 pretrained CLIP models on SUGARCREPE, with results reported in Table 6 and visualized in Figure 5.

**The best pretrained CLIP models demonstrate some compositional understanding but still have overall large rooms for improvements.** Table 6 shows that the largest pretrained CLIP models, *e.g.*, OpenAI's RN50x64, LAION's xlm-roberta-large-ViT-H-14, and DataComp's ViT-L-14, achieve near-human performance on REPLACE-OBJ. However, on REPLACE-OBJ, smaller models pretrained on small datasets still suffer from big drops in performance — 23% and 43% respectively for DataComp's small and medium models — compared to humans. Additionally, on nearly all other hard negative types, there are clear gaps (larger than 10%) between the best model performances and human performances, showing an overall large room for improvements in current models' compositionality.

---

[4]In Appendix D.3, we provide further results on training NEGCLIP with hard negatives filtered with our adversarial refinement mechanism.

Table 6: Our evaluation of pretrained CLIP models on SUGARCREPE shows that they demonstrate compositionality on some hard negatives but are far from human performance on others, especially on SWAP hard negatives or ones perturbing attributes and relations (also illustrated in Figure 5: lower overall performance on SWAP, and lower performances on attributes/relations compared to objects). We additionally evaluate recently introduced GPT-4V [33]. While it demonstrates strong results, there is still gap to human-level performance.

| Source | Model | Data Size | Model Size (M) | REPLACE | | | SWAP | | ADD | | Average |
|---|---|---|---|---|---|---|---|---|---|---|---|
| | | | | Object | Attribute | Relation | Object | Attribute | Object | Attribute | |
| | Human | | | 100 | 99 | 97 | 99 | 100 | 99 | 99 | 99 |
| Text-only model | Vera [27] | | | 49.39 | 49.62 | 49.36 | 49.19 | 49.40 | 49.42 | 49.57 | 49.42 |
| | Grammar [31] | | | 50.00 | 50.00 | 50.00 | 50.00 | 50.00 | 50.00 | 50.00 | 50.00 |
| OpenAI [36] | RN50 | 400M | 102 | 91.77 | 80.58 | 69.99 | 61.79 | 68.47 | 74.54 | 69.65 | 73.83 |
| | RN101 | | 120 | 92.49 | 83.88 | 67.07 | 56.50 | 65.92 | 75.46 | 70.09 | 73.06 |
| | ViT-B-32 | | 151 | 90.92 | 80.08 | 69.20 | 61.38 | 63.96 | 77.21 | 68.79 | 73.08 |
| | ViT-B-32-negclip | | 151 | 92.68 | 85.91 | 76.46 | 75.20 | 75.38 | 88.80 | 82.80 | 82.46 |
| | RN50x4 | | 178 | 92.68 | 82.99 | 67.57 | 65.04 | 63.36 | 79.34 | 70.09 | 74.44 |
| | RN50x16 | | 291 | 93.46 | 82.11 | 69.20 | 63.01 | 65.77 | 80.70 | 75.87 | 75.73 |
| | ViT-L-14 | | 428 | 94.07 | 79.19 | 65.15 | 60.16 | 62.31 | 78.32 | 71.53 | 72.96 |
| | RN50x64 | | 623 | 94.49 | 83.50 | 70.63 | 61.79 | 66.67 | 83.27 | 73.99 | 76.33 |
| LAION [42] | roberta-ViT-B-32 | 2B | 212 | 92.86 | 84.90 | 72.40 | 63.01 | 71.02 | 87.34 | 79.91 | 78.78 |
| | ViT-H-14 | | 986 | 96.49 | 84.77 | 71.76 | 67.48 | 73.12 | 92.05 | 85.84 | 81.64 |
| | ViT-g-14 | | 1367 | 95.76 | 85.03 | 72.40 | 63.01 | 71.17 | 91.51 | 82.08 | 80.14 |
| | ViT-bigG-14 | | 2540 | 96.67 | 88.07 | 74.75 | 62.20 | 74.92 | 92.19 | 84.54 | 81.91 |
| | xlm-roberta-base-ViT-B-32 | 5B | 366 | 93.16 | 84.01 | 69.20 | 63.41 | 67.57 | 87.78 | 81.07 | 78.03 |
| | xlm-roberta-large-ViT-H-14 | | 1193 | 96.85 | 86.04 | 72.05 | 63.82 | 72.07 | 93.11 | 86.13 | 81.44 |
| DataComp [14] | small:ViT-B-32 | 13M | 151 | 56.90 | 56.85 | 51.99 | 50.81 | 50.00 | 53.93 | 60.55 | 54.43 |
| | medium:ViT-B-32 | 128M | 151 | 77.00 | 69.54 | 57.68 | 57.72 | 57.06 | 66.73 | 64.88 | 64.37 |
| | large:ViT-B-16 | 1B | 150 | 92.68 | 79.82 | 63.94 | 56.10 | 57.66 | 84.34 | 78.61 | 73.31 |
| | xlarge:ViT-L-14 | 13B | 428 | 95.52 | 84.52 | 69.99 | 65.04 | 66.82 | 91.03 | 84.97 | 79.70 |
| OpenAI | GPT-4V [33] | | | 96.31 | 93.53 | 90.26 | 83.13 | 90.09 | 91.59 | 91.76 | 90.95 |

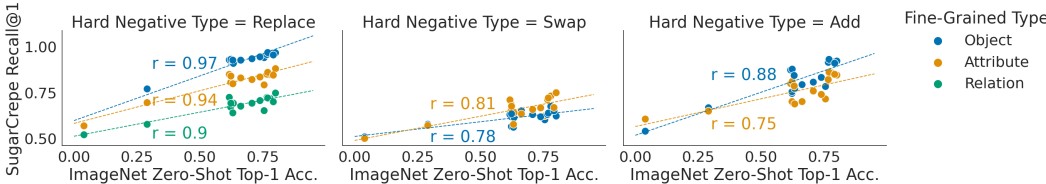

Figure 5: We plot pretrained vision-language models' zero-shot top-1 accuracy on ImageNet versus their retrieval recall@1 on SUGARCREPE, where $r$ is the Pearson correlation coefficient. This plot suggests that models' ImageNet zero-shot accuracy positively correlates with their compositionality.

**All models struggle at identifying SWAP hard negatives, regardless of their pertaining dataset and model size.** Among the three types of hard negatives, SWAP hard negatives present the biggest challenge to the pretrained CLIP models, even though humans can easily tell them apart from the positive captions. We observe in Table 6 that all models demonstrate low performance on both SWAP-OBJ and SWAP-ATT hard negatives regardless of their pretraining dataset and model sizes, with the difference from human performance reaching from 27% to 50%.

**Existing models are object-centric, struggling to compose attributes and relations.** We find that existing pretrained models are a lot better at composing objects than attributes or relations (Table 6). This finding holds for both REPLACE and ADD hard negatives but not the most difficult SWAP negatives, where models perform equally poorly on both SWAP-OBJ and SWAP-ATT. On REPLACE hard negatives, even though most models achieve human-level performance on REPLACE-OBJ, they all suffer from a drop in performance on REPLACE-ATT and REPLACE-REL, where the drop is as large as 15% and 29% respectively. Similarly, on ADD hard negatives, all models except for DataComp's small:ViT-B-32 experience a decrease in performance from ADD-OBJ to ADD-ATT, with the largest difference reaching 10%.

**Models' performance on SUGARCREPE correlates with their ImageNet zero-shot accuracy.** We show in Figure 5 that there is a positive correlation between models' performance on SUGARCREPE and their zero-shot accuracy on ImageNet. This correlation is moderate on SWAP-OBJ and ADD-ATT (Pearson correlation coefficient $r = 0.78$ and $r = 0.75$ respectively) and strong on all other hard negatives ($r > 0.8$).

# 6 Discussions

Our investigation reveals significant biases present in existing benchmarks for the compositional comprehension capability of vision-language models. The severity of this vulnerability is exemplified by text-only models without access to the image outperforming vision-language models. To address this, we introduce SUGARCREPE, a novel benchmark for evaluating the compositionality of vision-language understanding. Unlike previous benchmarks that relied on rule-based templates, we leverage large language models to generate less biased negatives and employ adversarial filtering mechanisms to minimize biases. Through reassessment of state-of-the-art models and recently proposed compositionality inducing mechanisms, we uncover a significant overestimation of their advancements, underscoring the need for further innovation.

## 6.1 Limitation and future work

**Scope of the compositionality benchmarks and vision-language models.** We focus our scope on compositionality benchmarks formulated as image-to-text retrieval task. While this is currently the most prevailing evaluation framework, future research can characterize compositionality evaluation as text-to-image retrieval problem, as in the initial efforts considered by [38, 45]. More importantly, we hope our work can guide future efforts in creating and ensuring faithful compositionality benchmarks in text-to-image form. In addition, we focus our evaluations on contrastively learned vision-language models [36]. Future work should include and characterize the compositionality of modern generative vision-language models [1, 7, 24, 46].

**Potential biases imposed by language models.** In this work, we identify *two* human interpretable dataset biases, the nonsensical and non-fluent biases, which may not cover all dataset artifacts that could possibly be exploited by a model. By leveraging ChatGPT in generating hard negatives, the generated captions may also exhibit hard to detect biases imposed by the language model, e.g., watermarks [21]. Future work may utilize more sophisticated adversarial filtering techniques that train models to detect and remove spurious dataset artifacts beyond human comprehension [51, 23].

**Shifts in language model behavior.** Our work leverages ChatGPT to generate hard negatives. However, recent work has pointed out that the underlying model behind these APIs may change, resulting in model behavior shifts [6, 28]. We discuss how this potential model behavior shift may affect our proposed dataset construction pipeline. Specifically, while there may be variances on the quality of the generated texts, we note that our employed adversarial refinement mechanism can ensure that the final evaluation set is free of the identified artifacts. In the case when ChatGPT improves and generates higher-quality captions, the refinement mechanism will filter out less examples and we can more efficiently create the final evaluation set. On the other hand, if ChatGPT degrades and shifts towards generating less fluent and plausible captions, the refinement mechanism will filter out more generated examples and we would need to generate more candidates in order to create an evaluation set of the same desired size. As a result, while the efficiency of the proposed dataset construction pipeline depends on quality of the language model used, our pipeline ensures the generated set does not contain the identified biases. In the large language model era, we see these capable models as productive tools one can leverage to efficiently process and create data. We do however deem careful validation mechanisms, such as our manual and automatic filtering technique, necessary to ensure that the ultimate goal is properly achieved.

## 6.2 Societal impact

As vision-language models such as CLIP [36] are becoming the foundation models for many downstream applications [40, 37], it is imperative to understand the limitations of these models to avoid misuses and undesirable outcomes [8, 2]. Compositionality benchmarks probe a model's understanding of finer-grained concepts, and hence allow us to identify blind spots [49, 53, 30] of seemingly powerful models deemed by standard classification and retrieval benchmarks [11, 26]. Our work further alleviates common artifacts in existing compositionality benchmarks that result in overestimation of a model's capability. We hope our proposed benchmark SUGARCREPE leads to more faithful assessment of a vision-language model's compositionality, and can hence guide more accurate usages of the models. Nevertheless, we note that strong performances on SUGARCREPE do not imply perfect models. We envision SUGARCREPE being one of the many benchmarks used to comprehensively understand the abilities of vision-language models from various aspects.

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

# A    Implementation details

## A.1    Hardware information

All experiments are run on a machine with an Intel(R) Xeon(R) CPU E5-2678 v3 with a 512G memory and two 48G NVIDIA RTX A6000 GPUs.

## A.2    Dataset sources

We obtain all existing datasets from their original sources released by the authors. We refer readers to these sources for the dataset licenses. To the best of our knowledge, the data we use does not contain personally identifiable information or offensive content.

- CREPE [30]: We obtain CREPE dataset from its official repository [5].
- ARO [49]: We obtain ARO dataset from its official repository [6].
- VL-CheckList [53]: We obtain VL-CheckList dataset from its official repository [7].
- COCO [26]: We obtain COCO from its official project website [8].

## A.3    Software configuration

**Models.** We detail the sources of the pretrained models we use in the paper, and the hyper-parameters used in training our own models.

- Vera model [27]: We obtain pretrained Vera model released by its author [9].
- Grammar model [31]: We obtain the Grammar model released by the authors [10].
- All pretrained CLIP models: We obtain all pretrained CLIP models' weights from Open-CLIP [11].
- NEGCLIP models: We obtain weights for pretrained NEGCLIP released by the authors [12]. For training from scratch and finetuning, we train RN50 and ViT-B/32 based on OpenCLIP codebase and set hyper-parameters as the following: number of warmup steps is 1000, batch size is 256, learning rate is 1e-4, weight decay is 0.1, number of epochs is 30. We augment the original CLIP loss with hard negative captions following NEGCLIP [49].

**Evaluations.** We base our evaluation framework on OpenCLIP [18]. We follow all default hyper-parameters used for evaluating models.

# B    Vision-language compositionality benchmarks

We provide an overview of existing vision-language compositionality benchmarks below, with Table 7 summarizing the dataset comparisons.

## B.1    Image-to-text formulation

A majority of current benchmarks formulate the evaluation task as image-to-text retrieval problem. These benchmarks generate hard negative texts procedurally through rule-based templates, where each benchmark considers different types of hard negatives.

**VL-Checklist [53].** VL-CheckList aims at evaluating vision-language models' understanding of different objects, attributes, and relationships. It contains REPLACE hard negatives generated by

---

[5] https://github.com/RAIVNLab/CREPE
[6] https://github.com/mertyg/vision-language-models-are-bows
[7] https://github.com/om-ai-lab/VL-CheckList
[8] https://cocodataset.org/
[9] https://huggingface.co/liujch1998/vera
[10] https://huggingface.co/textattack/distilbert-base-uncased-CoLA
[11] https://github.com/mlfoundations/open_clip
[12] https://github.com/mertyg/vision-language-models-are-bows

Table 7: Summary on vision-language compositionality benchmarks. SUGARCREPE considers image-to-text formulation to enable larger scale evaluation set. In addition, SUGARCREPE considers a wide range of hard negative types. SHUFFLE and NEGATE are omitted as they introduce inevitable biases discussed in Sec. 4.2.

| Benchmark | Task Formulation | Scale | Hard Negative Text Type | | | | |
| --- | --- | --- | --- | --- | --- | --- | --- |
| | | | SHUFFLE | REPLACE | SWAP | NEGATE | ADD |
| VL-CheckList [53] | Image-to-Text | > 1000 | | ✓ | | | |
| ARO [49] | Image-to-Text | > 1000 | ✓ | | ✓ | | |
| CREPE [30] | Image-to-Text | > 1000 | | ✓ | ✓ | ✓ | |
| Winoground [45] | Image-to-Text / Text-to-Image | 400 | | | ✓ | | |
| Cola [38] | Text-to-Image | 210 | | N/A | | | |
| SUGARCREPE | Image-to-Text | > 1000 | | ✓ | ✓ | | ✓ |

replacing atomic parts of the positive texts with other foils. VL-CheckList further breaks the hard negatives down into more granular categories based on the type of the replaced atomic part, *i.e.*, object, attribute, or relationship.

**ARO [49].** ARO focuses on models' understanding of different relationships, attributes, and order information. It considers SWAP and SHUFFLE hard negatives. SWAP hard negatives are generated by swapping two words in the positive texts; on the other hand, SHUFFLE hard negatives are generated by shuffling words in the positive texts. ARO further divides SWAP hard negatives into attribute or relationship type.

**CREPE [30].** CREPE is a large-scale evaluation benchmark that includes three types of hard negatives: REPLACE, SWAP and NEGATE. REPLACE and SWAP hard negatives are generated as in VL-CheckList and ARO. In addition, NEGATE hard negatives are generated by adding negation keywords (*i.e.*, *not* or *no*) to the original positive texts. The hard negatives are not further divided into fine-grained types (object, attribute, or relations).

## B.2 Text-to-image formulation

Complementary to image-to-text formulation, compositionality can as well be evaluated by probing a model to select an image that best matches a given text description, against other hard negative images as distractors. Unlike hard negative texts, hard negative images are more difficult to obtain and thus current text-to-image compositionality benchmarks are smaller at scale.

**Winoground [45].** Winoground is a small dataset manually curated by human annotators. Each example in the dataset contains two images and two matching captions, where both captions contain identical words that appear in different orders. Note that Winoground can be used for either image-to-text or text-to-image retrieval. While the original intention for Winoground is to evaluate vision-language compositionality, recent work [12] has pointed out that solving the tasks in Winoground requires not just compositional vision-language understanding, but additionally a suite of other abilities such as commonsense reasoning, or distinguishing visually difficult images.

**Cola [38].** Cola tests a vision-language model's ability to select an image that correctly matches a given caption, against another distractor image with the same objects and attributes but in the wrong composition. The image pairs are mined from existing datasets. As a result, the final evaluation set is relatively small in size (210 examples in total).

We deem text-to-image evaluation as important as image-to-text evaluation. Future work can explore approaches to generate or mine compositional hard negative images at scale, as preliminarily explored in [38, 49].

# C SUGARCREPE

## C.1 Taxonomy

Figure 6 shows the taxonomy of SUGARCREPE. We first categorize the hard negatives based on their forms: REPLACE, SWAP, and ADD. We then further divide each type of hard negatives into finer-grained sub-categories based on the type (object, attribute, or relation) of the atomic concept altered. SUGARCREPE covers a total of 7 fine-graind hard negative types.

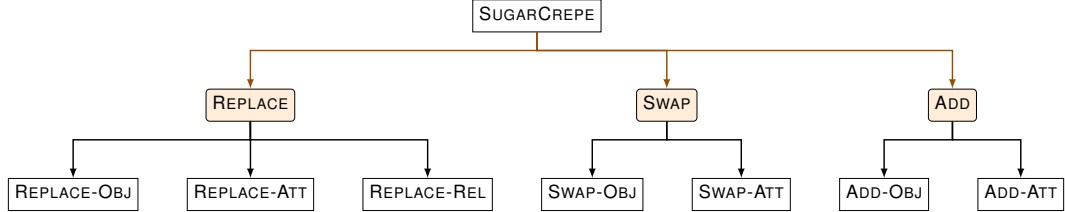

Figure 6: Taxonomy of hard negatives considered in SUGARCREPE.

## C.2 Hard negative generation procedure and templates

To generate hard negatives in SUGARCREPE, we come up with three different prompt templates for the three hard negative types considered: REPLACE, SWAP, and ADD. Each template consists of task instruction for generating the corresponding type of hard negatives and several (7 or more) few-shot demonstrations. We describe the general generation procedure and example prompt templates below and refer readers to our dataset repository for the full prompts used [13] .

**Generating REPLACE hard negatives.** To best leverage ChatGPT's capabilities, we devise a three-step workflow to generate REPLACE hard negatives: (1) We prompt ChatGPT in locating the desired atomic concepts (*e.g.*, objects) in the sentence; (2) We prompt ChatGPT to generate a new concept to replace a randomly selected old concept; (3) We let ChatGPT compose a new sentence by replacing the old concept with the new one. For steps (1) and (3), we prompt ChatGPT with a temperature of $0.0$ to get stable outputs. For step (2), however, we diversify the outputs by prompting ChatGPT with a higher temperature of $1.5$. With this design, we are able to generate diverse REPLACE hard negatives. Figure 7 shows the example templates and outputs for REPLACE hard negatives.

**Generating SWAP hard negatives.** To generate SWAP hard negatives, which do not require any new concepts, we simply prompt ChatGPT once with a temperature of 0.0. Unlike REPLACE, SWAP hard negatives are only possible when there are at least two atomic concepts of the same category, *i.e.*, either object or attribute. Thus, our prompt first queries ChatGPT whether it is possible to swap two atomic concepts in the input sentence to generate a new description. Only if the answer is yes, will ChatGPT then proceed to identify two swappable concepts and compose the corresponding new sentence by swapping the two concepts. Figure 8 shows the example templates and outputs for SWAP hard negatives.

**Generating ADD hard negatives.** Similar to the REPLACE, we also employ a three-step prompting procedure to generate ADD hard negatives. The only difference in the procedure is that we prompt ChatGPT to add the generated new concept to the original caption, instead of using it to replace an old concept. Figure 9 shows the example templates and outputs for ADD hard negatives.

## C.3 Adversarial refinement

We detail the adversarial refinement procedure below. Given a text model $M$, we denote its output score for the positive and negative caption of $i$-th image as $M(p_i)$ and $M(n_i)$. If $M(p_i) > M(n_i)$, then the model could identify the correct caption for the $i$-th image without referring to it. For a test set to be unattackable given the text model $M$, the expectation of $M$'s identifying the correct caption should be as close to random guess as possible; in particular, we hope that $E_i[M(p_i) > M(n_i)] = 0.5$. To achieve this for both the grammar model $M_1$ and plausibility model $M_2$, we first calculate the score

---
[13]https://github.com/RAIVNLab/sugar-crepe

```
Given an input sentence describing a scene, your task
is to:
1. Locate the noun words in the sentence.
2. Randomly pick one noun word.
3. Replace the selected noun word with a new noun word
to make a new sentence.

The new sentence must meet the following three
requirements:
1. The new sentence must be describing a scene that is
as different as possible from the original scene.
2. The new sentence must be fluent and grammatically
correct.
3. The new sentence must make logical sense.

Here are some examples:

Original sentence: A man is in a kitchen making pizzas.
Nouns: ["man", "kitchen", "pizzas"]
Selected noun: man
New noun: woman
New sentence: A woman is in a kitchen making pizzas.

Original sentence: a woman seated on wall and birds
besides her
Nouns: ['woman', 'wall', 'birds']
Selected noun: wall
New noun: bench
New sentence: A woman seated on a bench and birds
besides her.
```

```
Given an input sentence describing a scene, your task
is to:
1. Locate the adjective words describing objects in the
sentence. If there are no adjective words, return an
empty list.
2. Randomly pick one adjective word.
3. Replace the selected adjective word with a new
adjective word to make a new sentence.

The new sentence must meet the following three
requirements:
1. The new sentence must be describing a scene that is
as different as possible from the original scene.
2. The new sentence must be fluent and grammatically
correct.
3. The new sentence must make logical sense.

Here are some examples:

Original sentence: a blue bike parked on a side walk.
Adjectives: ["blue"]
Selected adjective: blue
New adjective: red
New sentence: a red bike parked on a side walk.

Original sentence: The kitchen is clean and ready for
us to see.
Adjectives: ["clean", "ready"]
Selected adjective: clean
New adjective: dirty
New sentence: The kitchen is dirty and ready for us to
see.
```

(a) REPLACE-OBJ.  (b) REPLACE-ATT.

```
Given an input sentence describing a scene, your task
is to:
1. Find any action or spatial relationships between two
objects in the sentence. If there are no such
relationships, return an empty list.
2. Randomly pick one relationship.
3. Replace the selected relationship with a new
relationship to make a new sentence.

The new sentence must meet the following three
requirements:
1. The new sentence must be describing a scene that is
as different as possible from the original scene.
2. The new sentence must be fluent and grammatically
correct.
3. The new sentence must make logical sense.

Here are some examples:

Original sentence: The dining table near the kitchen
has a bowl of fruit on it.
Relationships: ["near", "on"]
Selected relationship: near
New relationship: far from
New sentence: The dining table far from the kitchen has
a bowl of fruit on it.

Original sentence: A couple of buckets in a white room.
Relationships: ['in']
Selected relationship: in
New relationship: outside
New sentence: A couple of buckets outside a white room.
```

(c) REPLACE-REL.

Figure 7: Example prompt templates (black) and outputs (green) from ChatGPT for REPLACE hard negatives.

difference $g_i^{(1)} = M_1(p_i) - M_1(n_i)$ and $g_i^{(2)} = M_2(p_i) - M_2(n_i)$, where the range of both $g^{(1)}$ and $g^{(2)}$ is $[-1, 1]$. Then we split the 2D space of the joint range of $g^{(1)}$ and $g^{(2)}$ into $100 \times 100$ equal grids, and for each pair of symmetric grids, *e.g.*, $\{(g^{(1)}, g^{(2)}) | g^{(1)} \in (0.02, 0.04], g^{(2)} \in (-0.04, 0.06]\}$ and $\{(g^{(1)}, g^{(2)}) | g^{(1)} \in (-0.02, -0.04], g^{(2)} \in (0.04, -0.06]\}$, we preserve the same number of data for both grids, therefore we ensure that for the resultant set, $E_i[M_1(p_i) > M_1(n_i)] = 0.5$ and $E_i[M_2(p_i) > M_2(n_i)] = 0.5$.

## C.4 Dataset construction cost

We provide a high-level overview to the cost used to build SUGARCREPE by utilizing OpenAI's ChatGPT API for generating hard negatives. In building SUGARCREPE, we use approximately $40$ API calls to generate hard negatives for each COCO test caption, including all different fine-grained types of hard negatives. This amounts to a total of $25,000 \times 40 = 1,000,000$ API calls to ChatGPT. With each API call costing around $\$0.0005$, it took roughly $\$500$ to build SUGARCREPE.

```
Given an input sentence describing a scene, your task
is to first locate two swappable noun phrases in the
sentence, and then swap them to make a new sentence.
The new sentence must meet the following three
requirements:
1. The new sentence must be describing a different
scene from the input sentence.
2. The new sentence must be fluent and grammatically
correct.
3. The new sentence must make logical sense.

To complete the task, you should:
1. Answer the question of whether generating such a new
sentence is possible using Yes or No.
2. Output the swappable noun phrases.
3. Swap them to make a new sentence.

Here are some examples:

Input: A cat resting on a laptop next to a person.
Is it possible to swap noun phrases in the input
sentence to generate a new sentence that is different
from the input sentence and makes logical sense? Yes
Swappable noun phrases: laptop, person
Output: A cat resting on a person next to a laptop.

Input: A plate of donuts with a person in the
background.
Is it possible to swap noun phrases in the input
sentence to generate a new sentence that is different
from the input sentence and makes logical sense? Yes
Swappable noun phrases: a plate of donuts, a person
Output: A person with a plate of donuts in the
background.
```

(a) SWAP-OBJ.

```
Given an input sentence describing a scene, your task
is to first locate two swappable adjectives in the
sentence describing different objects, and then swap
them to make a new sentence.
The new sentence must meet the following three
requirements:
1. The new sentence must be describing a different
scene from the input sentence.
2. The new sentence must be fluent and grammatically
correct.
3. The new sentence must make logical sense.

To complete the task, you should:
1. Answer the question of whether generating such a new
sentence is possible using Yes or No.
2. Output the swappable adjectives.
3. Swap them to make a new sentence.

Here are some examples:

Input: A girl in a pink shirt holding a blue umbrella.
Is it possible to swap attributes in the input sentence
to generate a new sentence that is different from the
input sentence and makes logical sense? Yes
Swappable attributes: pink, blue
Output: A girl in a blue shirt holding a pink umbrella.

Input: A girl with a green shirt brushing her teeth
with a blue toothbrush.
Is it possible to swap attributes in the input sentence
to generate a new sentence that is different from the
input sentence and makes logical sense? Yes
Swappable attributes: green, blue
Output: A girl with a blue shirt brushing her teeth
with a green toothbrush.
```

(b) SWAP-ATT.

Figure 8: Example prompt templates (black) and outputs (green) from ChatGPT for SWAP hard negatives.

```
Given an input sentence describing a scene, your task
is:
1. Find the objects in the sentence.
2. Randomly pick one object.
3. Generate a new object that's not in the sentence.
4. Add the new object next to the selected object to
make a new sentence.

The new sentence must meet the following three
requirements:
1. The new sentence must describe a clearly new and
different scene.
2. The new sentence must be fluent and grammatically
correct.
3. The new sentence must make logical sense.

Here are some examples:

Original sentence: An elephant standing under the shade
of a tree.
Objects: ["elephant", "shade of a tree"]
Selected object: elephant
New object: squirrel
New sentence: An elephant and a squirrel standing under
the shade of a tree.

Original sentence: A bench at the beach next to the sea
Objects: ['bench', 'beach', 'sea']
Selected object: bench
New object: umbrella
New sentence: An umbrella and a bench at the beach next
to the sea.
```

(a) ADD-OBJ.

```
Given an input sentence describing a scene, your task
is:
1. Find the objects in the sentence.
2. Randomly pick one object.
3. Generate a new plausible but uncommon attribute for
this object that's not in the sentence.
4. Add the new attribute next to the selected object to
make a new sentence.

The new sentence must meet the following three
requirements:
1. The new sentence must describe a clearly new and
different scene.
2. The new sentence must be fluent and grammatically
correct.
3. The new sentence must make logical sense.

Here are some examples:

Original sentence: A large white airplane and a person
on a lot.
Objects: ["airplane", "person"]
Selected object: airplane
New attribute: blue
New sentence: A large white and blue airplane and a
person on a lot.

Original sentence: three people riding horses on a
beach
Objects: ['three people', 'horses', 'beach']
Selected object: three people
New attribute: elderly
New sentence: Three elderly people riding horses on a
beach.
```

(b) ADD-ATT.

Figure 9: Example prompt templates (black) and outputs (green) from ChatGPT for ADD hard negatives.

## C.5 Dataset information

We host SUGARCREPE on Github [14]. The data card [35] for SUGARCREPE, containing detailed dataset documentation, is available at the dataset repository [15]. We provide a summary below.

**Dataset documentation.** SUGARCREPE is a benchmark for faithful vision-language compositionality evaluation. Given an image, a model is required to select the positive text that correctly describes the image, against another hard negative text distractor that differs from the positive text only by small compositional changes. Each example consists of three fields:

---

[14] https://github.com/RAIVNLab/sugar-crepe
[15] https://github.com/RAIVNLab/sugar-crepe/blob/main/data_card.pdf

- `filename`: The id to an image
- `caption`: Positive text correctly describing the image
- `negative_caption`: Hard negative text incorrectly describing the image

**Maintenance plan.** We are committed to maintain the dataset to address any technical issues. We actively monitor issues in the repository.

**Licensing.** We license our work using MIT License [16]. All the source data we use is publicly released by prior work [26].

**Author statement.** We the authors will bear all responsibility in case of violation of rights.

# D Detailed evaluation results

## D.1 Full evaluation results on existing benchmarks

We provide the full evaluation results over 17 pretrained CLIP models as well as 2 text-only models, Vera [27] and the Grammar model [31], on existing compositionality benchmarks in Table 8. We see that the text-only models, arguably without any vision-language compositionality, outperform most of the pretrained CLIP models, achieving state-of-the-art performances on many benchmark tasks. This implies that current benchmarks fail to faithfully reflect a model's vision-language compositionality.

Table 8: Blind models (*i.e.*, Vera and Grammar model) outperform all 17 existing pretrained CLIP models on nearly all existing benchmark tasks. This implies that current benchmarks fail to faithfully measure a model's vision-language compositionality.

| Source | Model | CREPE | | | ARO | | | | VL-Checklist | | |
|---|---|---|---|---|---|---|---|---|---|---|---|
| | | Atomic | Swap | Negate | VG-Relation | VG-Attribution | COCO-Order | Flickr30K-Order | Object | Attribute | Relation |
| Text-only model | Vera [27] | 43.70 | 70.80 | 66.15 | 61.71 | 82.59 | 59.81 | 63.52 | 82.48 | 73.99 | 85.72 |
| | Grammar [31] | 18.15 | 50.88 | 9.77 | 59.55 | 58.38 | 74.33 | 76.26 | 57.95 | 52.35 | 68.50 |
| OpenAI [36] | RN50 | 26.47 | 28.32 | 31.25 | 53.87 | 63.37 | 44.89 | 52.46 | 86.85 | 68.30 | 75.95 |
| | RN101 | 27.63 | 32.74 | 12.50 | 52.43 | 62.93 | 29.86 | 39.34 | 86.44 | 67.93 | 71.75 |
| | RN50x4 | 26.24 | 28.32 | 9.51 | 51.59 | 62.27 | 29.39 | 34.56 | 87.23 | 68.74 | 73.81 |
| | ViT-B-32 | 22.31 | 26.55 | 28.78 | 51.12 | 61.33 | 37.14 | 47.18 | 87.00 | 68.80 | 77.04 |
| | RN50x16 | 26.36 | 29.65 | 9.38 | 52.13 | 62.71 | 29.95 | 34.26 | 86.95 | 69.34 | 76.83 |
| | RN50x64 | 26.82 | 30.09 | 23.57 | 51.00 | 62.56 | 40.54 | 46.74 | 87.71 | 68.61 | 74.97 |
| | ViT-L-14 | 26.36 | 25.66 | 24.74 | 53.34 | 61.50 | 36.11 | 45.08 | 87.86 | 68.27 | 75.89 |
| LAION [42] | ViT-H-14 | 23.70 | 25.22 | 16.54 | 50.33 | 62.93 | 25.79 | 30.96 | 85.39 | 68.46 | 71.13 |
| | ViT-g-14 | 23.70 | 24.78 | 20.70 | 51.60 | 61.20 | 25.59 | 30.10 | 86.07 | 69.43 | 71.03 |
| | ViT-bigG-14 | 23.58 | 24.78 | 17.97 | 51.61 | 61.89 | 25.24 | 30.22 | 84.66 | 67.80 | 66.48 |
| | roberta-ViT-B-32 | 22.66 | 21.24 | 20.31 | 47.46 | 62.00 | 24.77 | 30.76 | 85.71 | 68.82 | 65.90 |
| | xlm-roberta-base-ViT-B-32 | 21.16 | 20.80 | 12.76 | 47.93 | 59.73 | 23.85 | 30.32 | 86.06 | 70.41 | 63.01 |
| | xlm-roberta-large-ViT-H-14 | 24.16 | 23.89 | 20.05 | 46.14 | 57.84 | 26.05 | 31.00 | 87.89 | 70.25 | 63.89 |
| DataComp [14] | `small`:ViT-B-32 | 13.64 | 27.88 | 14.84 | 50.83 | 50.17 | 13.35 | 14.02 | 68.72 | 58.80 | 57.00 |
| | `medium`:ViT-B-32 | 16.42 | 20.35 | 11.33 | 50.45 | 54.04 | 16.44 | 16.26 | 78.43 | 63.53 | 62.94 |
| | `large`:ViT-B-16 | 18.15 | 17.26 | 17.06 | 48.82 | 53.21 | 21.49 | 26.44 | 84.73 | 65.72 | 64.81 |
| | `x-large`:ViT-L-14 | 21.62 | 22.57 | 16.28 | 48.54 | 60.03 | 23.19 | 29.52 | 86.66 | 67.01 | 67.93 |

## D.2 SUGARCREPE human evaluation

To compare the quality of the hard negatives generated in SUGARCREPE to those in current benchmarks (*i.e.*, ARO+CREPE), we randomly sample 100 examples for each of the hard negative types: REPLACE, SWAP, and NEGATE / ADD. Each example is organized to consist of (1) the original positive text, (2) its hard negative in ARO+CREPE, and (3) its hard negative in SUGARCREPE. For each example, a human user rates whether the hard negative in ARO+CREPE or that in SUGARCREPE is better (or tie) in terms of commonsense and grammatical correctness, respectively. Note that we compare NEGATE in ARO+CREPE to ADD in SUGARCREPE, as both hard negatives are intended to probe a model's understanding of the *existence or not* of an atomic concept. Table 9 shows that hard negatives in SUGARCREPE are much more sensical and fluent than that in ARO+CREPE across all three different types. For instance, SUGARCREPE has 68% more sensical and 46% more fluent hard negatives than ARO+CREPE on SWAP.

## D.3 Additional NEGCLIP results

In this section, we conduct preliminary experiments to answer whether models' performances on SUGARCREPE would increase hugely if the models are trained with hard negatives generated through

---

[16]https://github.com/RAIVNLab/sugar-crepe/blob/main/LICENSE

Table 9: Human evaluation results on the comparisons between hard negatives in ARO+CREPE and SUGARCREPE. We report the counts (out of 100 sampled examples) that the human user considers better or tie, w.r.t. both commonsense and grammatical correctness.

| Hard-negative Type | Evaluation | Human counts of better examples | | |
| --- | --- | --- | --- | --- |
| | | ARO+CREPE | SUGARCREPE | Tie |
| REPLACE | Commonsense | 11 | 29 | 60 |
| | Grammar | 4 | 33 | 63 |
| SWAP | Commonsense | 4 | 68 | 28 |
| | Grammar | 4 | 46 | 50 |
| NEGATE / ADD | Commonsense | 1 | 26 | 73 |
| | Grammar | 1 | 35 | 64 |

the same procedure as how we create SUGARCREPE. Since generating hard negatives for training with ChatGPT would incur substantial cost, we create hard negatives using a proxy method. In particular, we start with template-generated hard negatives on the COCO training set and apply our adversarial refinement technique to remove the biases. We use this adversarially refined dataset for NEGCLIP training. We show the results in Table 10. While we observe that the method improves over vanilla CLIP training without hard negatives, it performs similarly to NegCLIP and does not saturate the performance on SugarCrepe. This suggests that while the adversarial refinement mechanism prevents SugarCrepe from being attacked as an evaluation benchmark, leveraging the approach alone for training does not saturate the performance on SugarCrepe. Future work may characterize how LLMs could be used to generate better hard negatives for training to genuinely improve vision-language models' compositionality.

Table 10: Model performances on SUGARCREPE when trained with hard negatives generated through similar procedure as how SUGARCREPE is created.

| Model | Hard negative | SUGARCREPE | | |
| --- | --- | --- | --- | --- |
| | | REPLACE | SWAP | ADD |
| CLIP without hard negatives | N/A | 69.54 | 60.33 | 67.63 |
| NEGCLIP with template hard negatives | REPLACE | 74.32 | 62.65 | 72.92 |
| NEGCLIP with adversarial refined hard negatives | REPLACE | 73.37 | 61.40 | 72.84 |
| NEGCLIP with template hard negatives | SWAP | 73.31 | 68.35 | 71.93 |
| NEGCLIP with adversarial refined hard negatives | SWAP | 72.07 | 65.13 | 69.68 |
| NEGCLIP with template hard negatives | NEGATE | 72.74 | 60.89 | 70.47 |
| NEGCLIP with adversarial refined hard negatives | NEGATE | 72.70 | 60.75 | 68.70 |

