# A Limitation, future work, and societal impact

## A.1 Limitation and future work

There are several limitations to this work that future research can further explore. First, we focus our scope on compositionality benchmarks formulated as image-to-text retrieval task. While this is currently the most prevailing evaluation framework, future research can characterize compositionality evaluation as text-to-image retrieval problem, as in the initial efforts considered by [32, 39]. More importantly, we hope our work can guide future efforts in creating and ensuring faithful compositionality benchmarks in text-to-image form. Second, in this work, we identify *two* human interpretable dataset biases, the nonsensical and non-fluent biases, which may not cover all dataset artifacts that could possibly be exploited by a model. Future work may utilize more sophisticated techniques to remove spurious dataset artifacts beyond human comprehension [20]. Finally,