# OpenReview forum: "SugarCrepe: Fixing Hackable Benchmarks for Vision-Language Compositionality"
_NeurIPS.cc/2023/Track/Datasets_and_Benchmarks — NeurIPS 2023 Datasets and Benchmarks Poster_

### Official Review · Reviewer_bMrd · 2023-07-02
**A novel benchmark for evaluating compositionality of vision-language models**

**Rating:** 7
**Confidence:** 4
**Clarity:** The paper is well-written with suffic…

**Strengths:**

1. Authors choose to utilize language models to generate hard negatives with an adversarial refinement mechanism to refine synthesis. This approach is innovative and practical.
2. SUGARCREPE re-evaluates SOTA VLMs to show that the past improvements were overestimated, urging improvements in recently proposed compositionality-inducing strategies.
3. The paper is well-presented with clear details of prompts and textual operations for the proposed framework.

**Additional Feedback:**

In summary, with a clear presentation and reasonable responses during the rebuttal, I will rate it as an acceptance.

**Correctness:**

One worth-noting issue when constructing the dataset: Using a language model to generate hard negatives can lead to a lack of control over the exact nature of the negatives generated. While the paper mentions that they provide ChatGPT with specific prompts to guide the generation process, the generated output is not interpretable, and the evidence of generating such output can be unclear.

**Documentation:**

The dataset is well documented and is accessible through GitHub.

**Ethics:**

When generating hard negative texts, the authors should make sure that the output from the language model is inoffensive.

**Limitations:**

Yes, the authors have detailedly addressed the potential limitations of the work in the appendix.

**Opportunities For Improvement:**

1. More details on the computational cost of using language models should be included. The success of the approach will require a sufficient number of queries on ChatGPT. It will be helpful if authors evaluate the query cost (e.g., time of inference/API call), and discuss dataset effectiveness by the amount of queries.
2. Concrete definition and quantitative analysis of fluency and reasonableness for the generated hard negative will be beneficial to strengthen the motivation.
3. The quality of the hard negatives generated by SUGARCREPE is dependent on the capability of the language model used. Since OpenAI often adjusts/updates the model behind the ChatGPT interface, how does the work make the experiments consistent with this variable?
4. While the paper mentions that SUGARCREPE uses adversarial refinements to reduce biases, there is still a potential for textual bias imposed by the language model.

**Relation To Prior Work:**

The authors have compared the work with existing baselines.

**Summary And Contributions:**

The paper proposes SUGARCREPE to address the issue of biases in existing benchmarks for evaluating the compositional understanding of vision-language models (VLMs). The authors discovered that existing benchmarks impose artifacts when generating hard negatives. SUGARCREPE addresses this issue by using ChatGPT, instead of rule-based templates, to generate fluent and sensical hard negatives and utilizes an adversarial refinement mechanism to reduce biases.

---

> ### Author Response · Authors · 2023-08-19
> **Response to Reviewer bMrd**
>
> **Reviewer bMrd asked for the cost of using language models for constructing the dataset.** In building SugarCrepe, we use approximately 40 API calls to generate hard negatives for each COCO test caption, including all different fine-grained types of hard negatives. This amounts to a total of 25,000 * 40 = 1,000,000 API calls to ChatGPT. With each API call costing around 0.0005 USD, it took roughly 500 USD to build SugarCrepe. We have added this information to Appendix D4.
>
> **Reviewer bMrd asked for concrete definition and quantitative analysis of fluency and reasonableness.** We clarify that we quantitatively define fluency and reasonableness through grammar score and plausibility score respectively, as discussed in Sec. 3. In particular, we leverage a grammar check model and a plausibility estimation model to measure the fluency and reasonableness of a given caption. We show in Figure 1 that existing benchmarks exhibit biased score gap distributions for both fluency and reasonableness (where the distribution lies almost entirely on the positive spectrum), while the score gap distributions are balanced around zero in SugarCrepe (Figure 4). We clarified the definition in Sec. 3 in the revision.
>
> **Reviewer bMrd mentioned that the model behind ChatGPT may be updated. How do we make sure the quality of the generated hard negatives and the experiments are consistent.** This is a challenging open problem. In building SugarCrepe, we conduct manual filtering to verify the quality of the hard negatives generated by ChatGPT. We believe this manual verification stage can help make the dataset construction pipeline more robust to the change of ChatGPT’s model. More importantly, by open sourcing our dataset generation process and making the generated dataset publicly available, we ensure the experiments are consistent and reproducible.
>
> **Reviewer bMrd mentioned that there may be potential bias imposed by language models.** This is a great point, and we agree that there may still exist textual bias imposed by LLMs. In this work, we focus on removing two human interpretable biases: the non-fluent and non-plausible biases. We employ not only the adversarial refinement mechanism to reduce the biases, but also add a human filtering stage in the revision to ensure the validity of the generated hard negatives. To further remove potential textual biases that may be indistinguishable to humans, one future direction is to leverage adversarial filtering methods such as AFLite that train models to detect and filter out potential shortcuts in the examples [1,2]. We discuss this point in our limitation and future work section.
>
> [1] Rowan Zellers, Ari Holtzman, Yonatan Bisk, Ali Farhadi, Yejin Choi, “HellaSwag: Can a Machine Really Finish Your Sentence?”, 2019.
>
> [2] Ronan Le Bras, Swabha Swayamdipta, Chandra Bhagavatula, Rowan Zellers, Matthew E. Peters, Ashish Sabharwal, Yejin Choi, “Adversarial Filters of Dataset Biases”, in 2020.

---

> > ### Author Response · Authors · 2023-08-19
> > **Response to Reviewer bMrd (Cont.)**
> >
> > **Reviewer bMrd asked about the controllability and interpretability of the data generation process.** We agree that instructing LLMs to generate hard negatives may not be completely controllable as we eventually depend on the model’s creative writing capability to modify the positive captions. However, we employ multiple fine-grained mechanisms to control the generation pipeline and ensure the quality of the results. Specifically, we first carefully design prompt templates (Figure 7-9) to guide LLMs in generating negatives in a controllable and step-by-step way (detailed in Appendix D2 and illustrated through an example below). Then, we utilize various stages of automatic and manual filtering to ensure the validity of the generated negatives. Most importantly, we record all the intermediate and final results of the generated negatives throughout the entire generation and filtering process. As these traces consist of all textual outputs, we hope this maximizes the controllability and interpretability of the dataset generation process.
> >
> > We illustrate the generation pipeline using an example below. For instance, to generate a Replace-Obj negative for the caption “A man is in a kitchen making pizzas”, we first prompt the LLM to locate the objects in the caption, generating intermediate results of [“man”, “kitchen”, “pizzas”]. Then, we randomly select one of the located objects (e.g., “man”) using Python code, which we can explicitly control. We finally instruct the LLM to generate a replacement for the word “man” in the caption. We note that the LLM is considered more as an advanced sentence processing tool here for different purposes (locating objects, generating replacements) instead of a complete blackbox for text generation. After the negative is generated, we then go through automatic and manual filtering to check the validity of the example. We note that this “generate then filter” procedure resembles traditional crowdsourcing pipeline [3,4], where we leverage an LLM in the loop to increase the efficiency.
> >
> > [3] Olga Russakovsky, Jia Deng, Hao Su, Jonathan Krause, Sanjeev Satheesh, Sean Ma, Zhiheng Huang, Andrej Karpathy, Aditya Khosla, Michael Bernstein, Alexander C. Berg, Li Fei-Fei, “ImageNet Large Scale Visual Recognition Challenge”, 2014.
> >
> > [4] Ranjay Krishna, Yuke Zhu, Oliver Groth, Justin Johnson, Kenji Hata, Joshua Kravitz, Stephanie Chen, Yannis Kalantidis, Li-Jia Li, David A. Shamma, Michael S. Bernstein, Fei-Fei Li, “Visual Genome: Connecting Language and Vision Using Crowdsourced Dense Image Annotations”, 2016.

---

> > ### Comment · Reviewer_bMrd · 2023-08-20
> > **Concerns on the shift of base model behind API**
> >
> > Thank authors for providing convincing arguments on most of my concerns. However, regarding the base model behavior shift imposed by OpenAI (which is uncontrollable for most users), authors should have more solid arguments and quantitative evidence showing that manual verification of the generated texts can overcome this challenge. Note that, there are recent works [1,2] discussing this issue, the significant behavior shift behind the API, caused by model version control or knowledge shift.
> >
> > Given the current level of soundness of the rebuttal, I will keep my rating unchanged. Nevertheless, I am open to having further discussions on this intriguing issue. This concern would be a frequent challenge for researchers in the LLM era, and hence I hope that our discussion can be insightful to the community. I look forward to the authors' ideas and insights. Also, it will be beneficial for the paper to add some discussions/limitations on this concern.
> >
> > [1] Lingjiao Chen, Matei Zaharia, James Zou. *How is ChatGPT's behavior changing over time?*, 2023.
> >
> > [2] Yang Liu, Yuanshun Yao, Jean-Francois Ton, Xiaoying Zhang, Ruocheng Guo, Hao Cheng, Yegor Klochkov, Muhammad Faaiz Taufiq, Hang Li. *Trustworthy LLMs: a Survey and Guideline for Evaluating Large Language Models' Alignment*, 2023.

---

> > > ### Author Response · Authors · 2023-08-25
> > > **Further discussions on the shift of base model behind API**
> > >
> > > **Reviewer bMrd further discussed the challenge of potential model change of ChatGPT.** Thank you for opening further discussion on this intriguing issue. We added the suggested related work [1,2] and further discussion in our discussion/limitation section. In our context of using ChatGPT to generate hard negatives, we agree that potential model change behind ChatGPT can lead to variances on the quality of the generated texts (i.e., their false negative rate, fluency, and plausibility). Note that even the current ChatGPT is not perfect. However, despite the potential variances, our employed manual verification and adversarial refinement mechanism can still ensure that the final evaluation set is free of the identified artifacts. Specifically, in constructing SugarCrepe, manual verification is used to filter out 13% of false negative examples. More importantly, we observe that even ChatGPT can still generate less plausible captions such as “A mother elephant walks between the legs of its baby”. This observation thus motivated our further use of the proposed adversarial refinement mechanism, which is designed to filter out less fluent or plausible hard negatives. In our experiments, this adversarial refinement mechanism in fact filters out 57% of the ChatGPT-generated examples to ensure a balanced score distribution between the positive and the hard negative captions (as a comparison, it filters out 75% of the examples generated by template-based approaches). In the case when ChatGPT improves and generates higher-quality captions, the manual and automatic filtering rates will decrease and we can more efficiently create an evaluation set of the same size. If ChatGPT degrades and shifts towards generating lower-quality captions, the filtering rates will increase and we would need to generate more candidates in order to create a same-size final evaluation set. As a result, while the efficiency to create the dataset is subject to the quality of the LLM used, our pipeline ensures that the generated set does not contain the fluency and plausibility biases. In the LLM era, we see these capable models as productive tools one can leverage to process and create data. We do however deem careful validation, such as our manual and automatic filtering mechanisms, necessary to ensure the ultimate goal is properly achieved.
> > >
> > > [1] Lingjiao Chen, Matei Zaharia, James Zou. How is ChatGPT's behavior changing over time?, 2023.
> > >
> > > [2] Yang Liu, Yuanshun Yao, Jean-Francois Ton, Xiaoying Zhang, Ruocheng Guo, Hao Cheng, Yegor Klochkov, Muhammad Faaiz Taufiq, Hang Li. Trustworthy LLMs: a Survey and Guideline for Evaluating Large Language Models' Alignment, 2023.

---

> > > > ### Comment · Reviewer_bMrd · 2023-08-28
> > > > **Fair arguments with reasonable explanations**
> > > >
> > > > Thank the authors for continuing the discussion on the potential behavior shift. I appreciate the insightful response and raise the rating from 6 to 7.

---

### Official Review · Reviewer_6iRd · 2023-07-20
**Overall, interesting and relevant contribution to Neurips Datasets and Benchmarks.**

**Rating:** 7
**Confidence:** 3
**Correctness:** N/A

**Strengths:**

The paper observes an important limitation of existing compositional benchmarks. The paper is well organized, with empirical evidence motivating the work and convincing empirical results:
* The paper proposes strong evidence exhibiting the limits of existing evaluation benchmarks, in particular ARO and CREPE.
* The paper introduces a method to limit the biases in these datasets by generating hard negatives for image-to-text generation benchmark using LLMs.
* Using this method, the authors can re-evaluate the effectiveness of some approaches for compositional evaluation.

**Additional Feedback:**

N/A

**Clarity:**

The paper is very well written and easy to follow. Figures and Tables are overall very explicit and informative. There may be some minor fixes to consider:
* The name of Section 3 is maybe a bit misleading to me as we already introduced “current compositionally benchmarks”. Maybe just say “Limit and biases of current compositionally benchmarks”?
* There is a space missing in line 82 after reference [16].
* Figure 4, why is there no Shuffle line for SugareCrepe?
* Line 164, maybe use a comma separator for thousands: 12,164?
* It seems to me that lines 175 to 181 could be moved to Section 4.2. Otherwise, it is a bit difficult to understand how these 12,264 examples are generated and their relation with previous work.
* Section 5.1, I don’t really understand the metric used to compare the quality of hard negatives in SugareCrepe vs ATO + CREPE? What does this 35% represent?
* Table 4, what are the common sense grammar scores and where are they defined?
* Table 6, has a lot of figures, a bit difficult to read. Maybe add bold and underline figures for each section?
* Figure 5. Is it possible to define Recall @ 1 and top-1 accuracy?


**Documentation:**

Yes, there is sufficient detail to support reproducibility.

**Ethics:**

No, I do not suspect any ethical concerns with the submission that warrant further discussion or review.

**Limitations:**

Large language models (LLMs) are increasingly used to generate annotations. However, this raises a methodological question since the benchmarks used to evaluate these models, in part rely on these models. This raises the question about how sustainable and robust is the use of LLMs to generate evaluation data.

**Opportunities For Improvement:**

* It would be interesting to propose an update of Figure 2 but including this time on SugarCrepe, so we can directly visualize the impact of the method.
* Section 4.1. How do we explain that there are still some biases when generating captions with ChatGPT? Could you illustrate some biased examples?


**Relation To Prior Work:**

Relation to prior work is well explained. The related work section is very informative.


**Summary And Contributions:**

The paper observes an important limitation of existing compositional benchmarks. Many of the benchmarks can be solved by exploiting some structural biases regarding the plausibility and fluency of the examples. The paper proposes a procedure to generate hard negatives that allows to alleviate these biases. Using this procedure, they fix existing benchmarks and re-evaluate models and the new datasets. They show this leads to performance below expected, thus confirming the initial hypothesis that models could use the biases in the datasets to outperform them.

---

> ### Author Response · Authors · 2023-08-19
> **Response to Reviewer 6iRd**
>
> **Reviewer 6iRd asked to update Figure 2.** Thank you for the great suggestion. We will add SugarCrepe to Figure 2 for easier comparison.
>
> **Reviewer 6iRd asked us to provide examples of biased generations by ChatGPT.** While we design prompts to explicitly instruct ChatGPT in generating fluent and plausible hard negatives, there could still be cases where the generated captions are less plausible. For instance, given the positive caption “A baby elephant walks between the legs of its mother”, ChatGPT could generate a hard negative “A mother elephant walks between the legs of its baby”, which makes less sense. Similar examples include “A man on a surfboard riding a wave” with its generated hard negative “A wave is riding a man on a surfboard”. We thus employ the proposed adversarial refinement technique to further ensure the biases are minimized in the final evaluation set. Interestingly, we find ChatGPT can mostly (if not always) generate fluent sentences without problem. We will add such examples to the paper.
>
> **Reviewer 6iRd asked whether using LLMs to generate evaluation data is robust (reliable).** This is a good question. We agree that caution needs to be taken when using LLMs to generate evaluation data. To ensure the quality and the validity of the generated hard negatives, after the paper submission, we additionally employed a human filtering stage where we manually examine and filter out invalid hard negatives generated by ChatGPT. Specifically, as ChatGPT does not have access to the image, we manually check that the hard negatives are indeed describing different scenes to the original image. We see this semi-automatic dataset curation pipeline as an efficient way to scale compared to manual annotation from scratch, while the manual examination stage in our pipeline ensures the correctness and robustness of the created dataset. Our manual verification filtered out very few examples and did not change the results or conclusions of our paper.
>
> **Reviewer 6iRd suggested minor fixes on the presentation and asked clarifying questions.** Thank you for the suggestions. We will incorporate them in the revision and conduct a thorough round of polishing to further improve the presentation. We answer some clarifying questions below:
> - **Figure 4, why is there no Shuffle line for SugarCrepe?** We omit generating hard negatives by randomly shuffling the orders of words in the positive caption since that would almost always lead to non-fluent or non-plausible captions, which we aim to avoid.
>
> - **Section 5.1, I don’t really understand the metric used to compare the quality of hard negatives in SugarCrepe vs ATO + CREPE? What does this 35% represent?** Table 3 shows example comparisons between the hard negatives generated in SugarCrepe and those generated by template in ARO+CREPE. In Appendix E2, we further conducted a user study to verify that SugarCrepe contains more fluent and plausible hard negatives compared to those in ARO+CREPE. Particularly, given a tuple (original caption, SugarCrepe hard negative, ARO+CREPE hard negative), the user is asked to select which hard negative is more fluent and plausible: either from SugarCrepe, from ARO+CREPE, or tie. In the study, the user finds that the hard negatives from SugarCrepe are more fluent and plausible than that from ARO+CREPE on an average of 35% of examples.
>
> - **Table 4, what are the common sense grammar scores and where are they defined?** The grammar and the commonsense scores are the metrics we use to quantify the grammatical correctness and the plausibility of a given caption. Same as how we compute the score gap in Figure 1, the scores in Table 4 are defined as Grammar(T) and Vera(T), where Grammar(.) and Vera(.) is the grammar check and plausibility estimation model respectively. We will add clarification in the revision.

---

### Official Review · Reviewer_iLag · 2023-07-20
**A good work that points out a valid approach to construct unbiased visual compositional reasoning benchmarks**

**Rating:** 6
**Confidence:** 4
**Clarity:** Yes, the paper is well written and mo…

**Strengths:**

- This work successfully reveals what previous visual compositional reasoning benchmarks are tested on: some shortcuts, including grammatical patterns drawn from the template-based negative construction procedure, but not genuinely vision-language compositionality.

- The proposed method of leveraging large language models to generate candidate negatives is intuitive, easy to implement, and works well compared to templated-based approaches.

- Experimental results on CREPE show that the original version of the CREPE dataset has the hard negative problems mentioned. SUGARCREPE is a nice fix to this, meanwhile highlighting the difficulty of compositional visual reasoning.

**Additional Feedback:**

line 82: missing space before "For instance".

**Correctness:**

Yes, the dataset is constructed soundly. It also demonstrates what critical parts are missing for creating similar datasets in previous works.

**Documentation:**

Yes. The github repo seems valid and useful.

**Ethics:**

No foreseeable ethical concerns.

**Limitations:**

Yes, the author adequately addressed the limitations and potential negative societal impact of their work.

**Opportunities For Improvement:**

### Weaknesses

I think one major limitation of this work is the applicability of this method. Rewriting negative examples using large language models is a valid approach for fixing the existing shortcuts in CREPE. However, other vision-language compositional reasoning datasets do not show these shortcuts (e.g., Winoground's negatives are labeled by human participants and are not templated-based), so this work, SUGARCREPE, should likely be categorized as a bugfix for CREPE. Since it is a single-blind review process for this track, and the authors are from the same lab that creates CREPE. I suggest the authors can release a bug-fixed version of the original CREPE (which appeared in CVPR 2023) and not build a separate benchmark for this.

I would like to see more analysis of existing vision-language compositional reasoning datasets beyond CREPE since somehow I think the hard negative problem only exists in CREPE, and such LLM-based rewriting method may not be effective for other similar benchmarks.

**Relation To Prior Work:**

Yes, the authors clearly discussed how this work differs from previous contributions.

**Summary And Contributions:**

This work investigates the topic of vision-language compositional reasoning by diving into existing benchmarks and analyzing how the negative examples were built. The authors find that existing hard negatives can lead to potential shortcuts for evaluating vision-language models. Thus they cannot capture the compositionality component as expected. To solve this, this paper proposes utilizing large language models to generate fluent and reasonable hard negatives, easing the biased in the original vision-language compositional reasoning datasets. Experiments on CREPE with CLIP models show that the negatives do affect models' performance to some extent, and models still struggle to reach satisfactory compositional reasoning capabilities.

---

> ### Author Response · Authors · 2023-08-19
> **Response to Reviewer iLag**
>
> **Reviewer iLag asked about the general applicability of the proposed dataset construction method.** There appears to be a confusion about our work. We have identified biases not only in CREPE but a number of other compositional benchmarks, such as ARO and VL-CheckList [1,2]. We propose a method for generating faithful compositionality benchmarks, it can be generally applied to fix the bias problem in all these existing benchmarks. As such, while our work shares the similarity in naming to CREPE, it isn’t really a “bug fix” only for CREPE as it is an improvement that can be applied to fix all existing benchmarks with biases. Indeed, we agree that benchmarks like Winoground with human-crafted negatives may not be subject to the identified biases. We consider the proposed dataset construction pipeline as an alternative, scalable way to curate large compositionality benchmarks efficiently compared to fully utilizing manual annotation. We provide more detailed discussions on more existing compositionality benchmarks including Winoground in Appendix C.
>
> **Reviewer iLag asked whether the bias problem only exists in CREPE and would like to see more analysis on existing benchmarks.** We would like to highlight that the bias problem we identified exists not only in CREPE but also in other popular compositionality benchmarks: ARO and VL-CheckList. All these template-generated benchmarks are increasingly being used in the literature, and our analysis in Sec. 3 shows the severe bias problem exists in all CREPE, ARO, and VL-CheckList. In the experiments, we extensively evaluate all models on CREPE, ARO, VL-CheckList and SugarCrepe, where we show that our proposed dataset construction approach effectively mitigates the biases common in all existing benchmarks: First, we see that shortcuts originally in all template-generated benchmarks no longer exist in SugarCrepe by seeing blind models’ performances become nearly random. Furthermore, we reveal the existence of shortcuts in current benchmarks (CREPE, ARO, and VL-CheckList) leads to overestimation of NegCLIP’s effectiveness, where its performance on SugarCrepe is comparably small when the shortcuts are removed.
>
> [1] Mert Yuksekgonul, Federico Bianchi, Pratyusha Kalluri, Dan Jurafsky, James Zou, “When and Why Vision-Language Models Behave like Bags-Of-Words, and What to Do About It?”, 2023.
>
> [2] Tiancheng Zhao, Tianqi Zhang, Mingwei Zhu, Haozhan Shen, Kyusong Lee, Xiaopeng Lu, Jianwei Yin, “VL-CheckList: Evaluating Pre-trained Vision-Language Models with Objects, Attributes and Relations”, 2022.

---

> > ### Comment · Reviewer_iLag · 2023-08-28
> >
> > Thanks for the author's response. I have no further questions. My score remains the same.

---

### Official Review · Reviewer_eSt3 · 2023-07-21
**Well-motivated, Novel, and Impactful New Benchmark**

**Rating:** 9
**Confidence:** 4
**Clarity:** The paper is well-written and easy to…

**Strengths:**

1. The paper is well-motivated by the finding of biases in existing benchmarks (Section 3). Quantitative results of blind models give solid evidence of such biases (Section 3).
2. The proposed method to construct more plausible and fluent hard negatives is well-suited to address the bias problem.
3. The evaluation of existing vision-language models (e.g., NegCLIP (Section 5.2) and 17 CLIP models (Section 5.3)) on the new benchmark is extensive and comprehensive. The experiments also show good insights into which specific types of hard negative models suffer more compositionality problems.
4. The paper is well-written and easy to follow.

**Additional Feedback:**

L111: The formal name of “OpenAI in-house data” is WebImageText (WIT).

L132: in-context learning: cite [D]

**References**

[D] Tom B. Brown, Benjamin Mann, Nick Ryder, Melanie Subbiah, Jared Kaplan, Prafulla Dhariwal, Arvind Neelakantan, Pranav Shyam, et al., “Language Models are Few-Shot Learners,” in NeurIPS, 2020.

**Correctness:**

The claims made in the submission are correct. The dataset is well-motivated by the flaws in previous datasets, and the bias problem is well-addressed by the proposed LLM-based approach and adversarial filtering.

**Documentation:**

Sufficient details on data generation based on LLM and adversarial filtering are given. The URL to the dataset (GitHub repo) is provided.

**Ethics:**

There are no ethical concerns with this submission.

**Limitations:**

The paper has adequately addressed the limitations and potential negative societal impact (Appendix A).

**Opportunities For Improvement:**

**[Simple Fix with LLM-based Hard Negatives]** I have one concern that this new benchmark can be quickly saturated by methods using large language models (LLM) to generate hard negatives (similar to the proposed approach for building the benchmark) for training. If possible, can the authors provide the results of this new baseline?

**[Bias may also exist in LLM-generated content]** While the paper is motivated by the non-plausible and non-fluent bias introduced by straightforward text augmentation, texts generated by LLM may also contain signals for blind models to detect whether they are generated or not, especially due to the recent efforts of watermarking in LLM (e.g., [A]). I don’t expect this problem can be solved in this submission, but do expect the authors to add this to the limitation section.

**References**

[A] John Kirchenbauer, Jonas Geiping, Yuxin Wen, Jonathan Katz, Ian Miers, and Tom Goldstein, “A Watermark for Large Language Models,” in ICML, 2023.

**Relation To Prior Work:**

The paper has clearly discussed the contribution compared to previous benchmarks. One small suggestion is to add a discussion on a similar problem in the related compositional text-to-image generation problem [B].

**References**

[B] Dong Huk Park, Samaneh Azadi, Xihui Liu, Trevor Darrell, and Anna Rohrbach, “Benchmark for Compositional Text-to-Image Synthesis,” in NeurIPS Datasets and Benchmarks Track, 2021.

**Summary And Contributions:**

The paper first finds that existing compositional vision-language understanding benchmarks are hackable by containing non-plausible and non-fluent biases, which even enables language-only models to achieve state-of-the-art performances. To address this problem, the paper introduces SugarCrepe, a new benchmark to evaluate vision-language compositionality. Concretely, to address the non-plausible and non-fluent biases in the existing hard negative generation process, the paper leverages large language models to generate plausible and fluent hard negatives followed by adversarial filtering. By re-evaluating existing methods on the SugarCrepe, the paper finds that existing methods’ compositionality is overestimated when evaluated on previous hackable benchmarks. Furthermore, the results of state-of-the-art CLIP models show that they still lack compositionality.

---

> ### Author Response · Authors · 2023-08-19
> **Response to Reviewer eSt3**
>
> **Reviewer eSt3 asked whether models trained with hard negatives would saturate SugarCrepe, where the hard negatives are generated using our approach.** This is an interesting question and one that took us some amount of effort to answer this week. We generated a training set with hard negatives and trained a few CLIP models with it. To create SugarCrepe, we had used ChatGPT to generate the hard negatives. While this was affordable to do for an evaluation set, it becomes financially irresponsible for us to create a training set with its current API cost  (estimated to be $11,500 to create a training set the size of MSCOCO’s training set). Therefore, we created hard negatives this past week using a proxy method and evaluated its performance on SugarCrepe. In particular, we start with template-generated hard negatives on the MSCOCO training set and apply our adversarial refinement technique to remove the biases. We use this adversarially refined dataset for training.
>
> We provide the results in the following table. While we observe that the method improves over vanilla CLIP training without hard negatives, it performs similarly to NegCLIP and does not saturate the performance on SugarCrepe. This suggests that while the adversarial refinement mechanism prevents SugarCrepe from being attacked as an evaluation benchmark, leveraging the approach alone for training does not saturate the performance on SugarCrepe. Future work may characterize how LLMs could be used to generate better hard negatives for training to genuinely improve vision-language models’ compositionality.
>
> | Evaluation on SugarCrepe    | Replace |  Swap  |   Add  |
> |-----------------------------|:-------:|:------:|:------:|
> | CLIP without hard negatives |  69.54  |  60.33 |  67.63 |
> | NegCLIP (Replace)           |  74.32  |  62.65 |  72.92 |
> | New baseline (Replace)      |  73.37  |  61.40 |  72.84 |
> | NegCLIP (Swap)              |  73.31  |  68.35 |  71.93 |
> | New baseline (Swap)         |  72.07  |  65.13 |  69.68 |
> | NegCLIP (Negate)            |  72.74  |  60.89 |  70.47 |
> | New baseline (Negate)       |  72.70  | 60.75  |  68.70 |
>
> **Reviewer eSt3 points out LLM-generated content may exhibit hard to detect invisible watermarks.** This is a great point. We agree that LLM-generated texts may have biases that we haven’t considered. Future work could leverage adversarial filtering methods that train models to detect and filter out potential shortcuts in the datasets [1,2]. We have mentioned a note about this in our limitation and future work section.
>
> **Reviewer eSt3 asked to comment on a similar bias problem in related compositional text-to-image generation problem [3].** In compositional text-to-image generation problem, it is indeed possible that the synthesized captions exhibit biases (e.g., grammatical errors or that the synthesized captions are longer), which may obstruct correct interpretation of a model’s performance on synthesized captions as compared to natural captions. We are very glad to see that the original authors had taken this problem into account where they aim to generate synthesized captions by maintaining similar sentence structures to natural captions to avoid potential artifacts [3]. We added this related work in the revision.
>
> [1] Rowan Zellers, Ari Holtzman, Yonatan Bisk, Ali Farhadi, Yejin Choi, “HellaSwag: Can a Machine Really Finish Your Sentence?”, in ACL 2019.
>
> [2] Ronan Le Bras, Swabha Swayamdipta, Chandra Bhagavatula, Rowan Zellers, Matthew E. Peters, Ashish Sabharwal, Yejin Choi, “Adversarial Filters of Dataset Biases”, in ICML 2020.
>
> [3] Dong Huk Park, Samaneh Azadi, Xihui Liu, Trevor Darrell, and Anna Rohrbach, “Benchmark for Compositional Text-to-Image Synthesis,” in NeurIPS Datasets and Benchmarks Track, 2021.

---

> > ### Comment · Reviewer_eSt3 · 2023-08-28
> >
> > I have read other reviewers' comments and authors' responses. The response clears all of my concerns. I raise my rating to "strong accept."

---

### Author Response · Authors · 2023-08-19
**General Response**

**General Response:** We thank all reviewers for their positive comments and valuable suggestions. We thank Reviewer eSt3 for finding our paper “well-motivated with solid evidence”, our approach “well-suited”, and that our evaluation is “extensive and comprehensive”. We thank Reviewer iLag for finding that our work “successfully reveals the shortcuts in previous benchmarks'', the proposed method “intuitive, easy to implement, and works well”, and our experimental results “highlighting the difficulty of compositional visual reasoning”. We thank Reviewer 6iRd for mentioning that our paper “observes an important limitation of existing benchmarks with empirical evidence motivating the work and convincing empirical results''. We thank Reviewer bMrd for finding our proposed approach “innovative and practical”, and that our paper is “well-presented with clear details”.

In this response, we discuss additional improvements made to ensure the quality of the proposed dataset, discuss additional experimental results and the revision added to our manuscript; we also answer clarifying questions to individual reviewer’s comments. Please let us know if there are any follow-up questions, and we will make sure to answer all of them during the paper discussion period and in our final revision.

**Reviewer eSt3 and bMrd asked about the potential existence of biases in LLM-generated texts.** We thank the reviewers for noting there may be potential biases in LLM-generated text. To mitigate some of these biases, we employ manual filtering to ensure the validity of LLM-generated hard negatives. Specifically, one potential bias in LLM-generated texts is that the generated caption may be “false negative”. For instance, given an image with a positive caption “a man and a child sitting on a sofa”, a compositional change that replaces “child” with “girl” may still result in a correct caption. In the revision, we address this bias by conducting a thorough round of manual verification to filter out false negative captions, where the false negative rate is around 13%. We added a discussion and revised the manuscript to highlight this step. We find that the key takeaways in the experiments remain the same whether we do the manual verification, mainly because there are few such false negatives overall. In our released dataset repository, we provide both human-filtered and non-filtered versions of SugarCrepe for transparency and reproducibility. Finally, we added discussions on how future work could leverage general adversarial filtering techniques [1,2] to remove potential LLM-generated artifacts that are beyond human comprehension.

**Reviewer eSt3 wonders whether performance on SugarCrepe would saturate if models are trained with hard negatives, where the hard negatives would be generated using our approach.** We conducted additional experiments to answer this point. As using ChatGPT to generate hard negatives for a large training set takes a long time with significant financial cost (e.g., $11,500 estimated for curating a training set the size of MSCOCO’s training set), we evaluate with a proxy method where we instead start with template-generated hard negatives and apply adversarial refinement to remove the biases. We find this approach leads to better performance compared to vanilla CLIP training without hard negatives, but does not saturate the performance on SugarCrepe. We provide the results and more detailed discussions in the response to Reviewer eSt3 Question 1.

[1] Rowan Zellers, Ari Holtzman, Yonatan Bisk, Ali Farhadi, Yejin Choi, “HellaSwag: Can a Machine Really Finish Your Sentence?”, 2019.

[2] Ronan Le Bras, Swabha Swayamdipta, Chandra Bhagavatula, Rowan Zellers, Matthew E. Peters, Ashish Sabharwal, Yejin Choi, “Adversarial Filters of Dataset Biases”, 2020.

---

### Decision · Program_Chairs · 2023-09-22

**Decision:**

Accept (Poster)

**Comment:**

This work points out that existing compositional vision-language understanding benchmarks are hackable by containing non-plausible and non-fluent biases, which even enables language-only models to achieve state-of-the-art performances. Then the authors introduce SugarCrepe, a new benchmark to evaluate vision-language compositionality. The authors leverage large language models to generate plausible and fluent hard negatives followed by adversarial filtering. By re-evaluating existing methods on the SugarCrepe, the paper finds that existing methods’ compositionality is overestimated when evaluated on previous hackable benchmarks. This work provides a challenging benchmark and interesting findings. Please make sure to add the promised results and related discussion in the final revision.